



# Modeling of the large-scale nutrient biogeochemical cycles in Lake Onego

Oleg P. Savchuk[1], Alexey V. Isaev[2,3], Nikolay N. Filatov[2,3]

[1] Baltic Nest Institute, Stockholm University Baltic Sea Centre, Stockholm, 10691, Sweden

[2] Shirshov Institute of Oceanology, Russian Academy of Sciences, Moscow, 117997, Russia

[3] Northern Water Problems Institute, Karelian Research Center, Russian Academy of Sciences, Petrozavodsk, 185030, Russia

*Correspondence to*: Oleg Savchuk (oleg.savchuk@su.se), Alexey Isaev(isaev.av@spb.ocean.ru)

**Abstract.** Despite a long history of research, there is almost no information regarding the major biogeochemical fluxes that could characterize the past and present state of the European Lake Onego ecosystem and be used for reliable prognostic

estimates of its future. To enable such capacity, we adapted and implemented a three-dimensional coupled hydrodynamical biogeochemical model of the nutrient cycles in Lake Onego. The model was used to reconstruct three decades of Lake Onego ecosystem dynamics with daily resolution on a $2 \times 2$ km grid. A comparison of available information from Lake Onego and other large boreal lakes proves that this hindcast is plausible enough to be used as a form of reanalysis. As new regional phenological knowledge, the reanalysis quantifies that the spring phytoplankton bloom, previously overlooked, reaches a

maximum of $500 \pm 128$ mg C m$^{-2}$ d$^{-1}$ in May, contributes to approximately half of the lake's annual primary production of $17.0$–$20.6$ g C m$^{-2}$ yr$^{-1}$, and is triggered by increasing light availability rather than by an insignificant rise in water temperature. Coherent nutrient budgets provide reliable estimates of phosphorus and nitrogen residence times of 47 and 17 years, respectively. The shorter nitrogen residence time is explained by sediment denitrification, which in Lake Onego removes over 90% of the bioavailable nitrogen input, but is often ignored in studies of other large lakes. This model can be used for long-

term projections as soon as the corresponding scenarios of climate change and socio-economic development become available for north-western Russia.

## 1 Introduction

Being situated within the Baltic Sea drainage basin, boreal Lake Onego is the second-largest lake in Europe. Lake Onego is strongly phosphorus limited and still largely oligotrophic due to minor anthropogenic influence (Rukhovets and Filatov, 2010;

Filatov and Rukhovets, 2012; Kalinkina et al., 2020). Total phosphorus (TP) inputs to Lake Onego, expressed either as a yield from the catchment area (15 kg TP km$^{-2}$ yr$^{-1}$) or per unit of the lake's water volume (3 mg TP m$^{-3}$ yr$^{-1}$), are similar to those in boreal Lake Ladoga (14 kg TP km$^{-2}$ yr$^{-1}$; 4 mg TP m$^{-3}$ yr$^{-1}$, respectively), the largest lake in Europe. For comparison, the net anthropogenic input of phosphorus generated at the Great American Lakes watershed has decreased over time from 21 kg TP km$^{-2}$ yr$^{-1}$ in 1987 to 10 kg TP km$^{-2}$ yr$^{-1}$ in 2012 (Howarth et al., 2021). The future evolution of the Lake Onego ecosystem may

be driven by both the existing registered warming in its catchment area (Käyhkö et al., 2015; Filatov et al., 2018; 2019) and the projected effects of climate change (Saraiva et al., 2019; Meier et al., 2019), as well as socio-economic development (Zandersen et al., 2019; Okrepilov et al., 2020; Pihlainen et al., 2020). Current projections indicate that river discharge will increase in the northern Baltic Sea region, subsequently increasing waterborne nutrient inputs (BACC II Author Team, 2015) and mobilizing phosphorus reserves accumulated in the drainage area (McCrackin et al., 2018). The interaction of these

regionally occurring changes with the anticipated augmentation of local industrial, agricultural, mining, and forestry activities (Bartosova et al., 2019), including aquaculture (Sterligova et al., 2018), could generate synergetic ecosystem effects, such as those already occurring in the nutrient-rich Lake Winnipeg (Schindler et al., 2012). Hence, we need the capacity and a tool to reliably describe the current state of Lake Onego and to make prognostic estimates of future scenarios.



The development of mathematical models of large lake ecosystems and their implementation for producing recommendations on environmental protection measures has been ongoing for several decades (e.g., DiToro and Connoly, 1982; Strashkraba and Gnauk, 1985; Mooij et al., 2010; Zhang et al., 2013; Bocaniov et al., 2016; Scavia et al., 2016; Vinçon-Leite and Casenave, 2019), including modeling of Ladoga and Onego lakes (Menshutkin and Vorobyeva, 1987; Astrakhantsev et al., 1996; Menshutkin et al., 1998; Rukhovets et al., 2003; Rukhovets and Filatov, 2010; Filatov, 2020; Isaev and Savchuk, 2020). The main attention in Lake Ladoga and Onego modeling has been focused on spatial–temporal changes in nutrient concentrations, while the mechanism of these changes, although reproduced by the models as biogeochemical transports and transformations, remained largely hidden and unanalyzed. Another important deficiency in prevailing modeling was the neglect of the sediments with its nutrient dynamics, which serve as a "memory" of the lake ecosystem's evolution and an important link that closes the biogeochemical cycles through the remineralization of nutrients. In the present study, we adapted and implemented the St. Petersburg Model of the Baltic Sea Eutrophication (SPBEM) with a particular focus on the analysis of the biogeochemical fluxes in Lake Onego, including its bottom sediments. Importantly for the long-term forecasting, SPBEM describes the coupled cycles of nitrogen, phosphorus, and silica, which enables its use in water bodies with spatially and temporally changing limitations by any of these nutrients. A good illustration of the SPBEM performance in waters with variable nutrient limitations is the reliable reproduction of the spatial gradients of limiting nutrients in the Neva River estuary in the easternmost Gulf of Finland (Isaev et al., 2020).

Despite wide-ranging research on the Lake Onego ecosystem (e.g., Rukhovets and Filatov, 2010; Filatov et al., 2018; 2019; Kalinkina et al., 2020; Filatov, 2020), the drivers and components of Onego's biogeochemical cycles are still understudied. There is neither quantitative knowledge nor field data, especially on major biogeochemical fluxes, that could characterize the past and present of the Lake Onego ecosystem and could be used for reliable prognostic estimates of its future. Therefore, we implemented our ecosystem model, validated as extensively as the available data permitted, for the reconstruction of three decades of Lake Onego ecosystem dynamics with daily resolution. We also verified the model's consistency by analyzing and demonstrating how it reproduced the major mechanisms that determine seasonal dynamics. The simulated three-dimensional fields of both pelagic and sediment variables, as well as the fields of most important biogeochemical fluxes, can be considered as a form of "biogeochemical reanalysis", albeit without formal data assimilation. To the best of our knowledge, this is the first use of a three-dimensional coupled hydrodynamic biogeochemical model to reconstruct past long-term biogeochemical dynamics in a large boreal lake, which presents new knowledge about the Lake Onego ecosystem. Although our model reproduces a limited number of components at the lower levels of lake ecosystem, simulated variables and fluxes can be used with existing knowledge on the relationships between other components, for instance, between the nutrient dynamics in sediments and benthic communities (e.g., Kalinkina and Belkina, 2018; Ehrnsten et al., 2020).

Reanalysis with data assimilation has recently been implemented for short-term to long-term reconstructions of oceanic and marine ecosystems, with differing degrees of success (e.g., Ciavatta et al., 2016; Liu et al., 2017; Fennel et al., 2019, Kõuts et al., in press). The most important weakness of the data assimilation algorithm used in biogeochemical models based on mass balance principles is its inherent non-conservativeness. Consequently, the biogeochemical model, being reasonably calibrated during hindcast with data assimilation, may display deviant behavior in forecasting when left unsupported by data. The avoidance of such biases is especially important for our intention of producing prognostic estimates when corresponding scenarios of climate change and socio-economic development become available for northwest Russia.



## 2 Model and data

### 2.1 Model presentation

SPBEM is a coupled 3D hydrodynamical–biogeochemical model that performed reasonably well in hindcasts and scenarios of climatic changes and nutrient load reductions for the Baltic Sea (e.g., Ryabchenko et al., 2016; Meier et al., 2018, 2019).

SPBEM consists of biogeochemical and hydrodynamical modules. The former simulates the biogeochemical cycles of nitrogen, phosphorus, and silicon in the water column and bottom sediments. One of the technical advantages of using this model for the Lake Onego ecosystem is its capability to process and display not only indicators of trophic state (concentrations), but also the biogeochemical fluxes that determine the temporal and spatial dynamics of nutrient concentrations. All equations, parameterizations, coefficients, and constants of the biogeochemical module of SPBEM are

presented in full detail, enabling its independent reproduction and implementation for other water bodies, by Isaev et al. (2020). To avoid the extensive self-citation, we are not repeating those our formulations in the present paper.

The adaptation of the biogeochemical module of the SPBEM model to Lake Onego conditions consisted of redefining the autotroph variables and processes introduced in the original model (Savchuk, 2002, Isaev et al., 2020). Assuming the negligibility of nitrogen fixation in severely phosphorus-limited lake, we excluded both the process and its performers, the

diazotrophic cyanobacteria functional group, from the implemented formulation. Thus, autotrophs were presented by only two variables, diatoms and non-diatoms, that comprised all the other (summer) phytoplankton species, for example, chlorophytes, chrysophytes, and cyanobacteria. Besides, we omitted the silicon cycle from the model because, according to the occasional observations (Sabylina et al., 2010; Efremova et al., 2019; Ryzhakov et al., 2019), silicate concentration during summer and autumn development of diatoms ranged within 200–400 mg Si m$^{-3}$ even in the deep-water areas, that is never became limiting.

The modified scheme of the biogeochemical interactions is shown in Fig. 1.

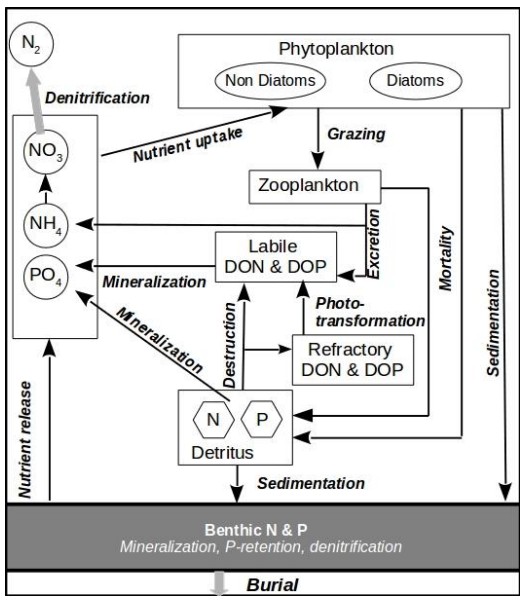

**Figure 1: Simplified presentation of the SPBEM model variables and nutrient fluxes.**

The hydrodynamical module was built on the University of Massachusetts MITgcm model (Marshall et al., 1997a, 1997b). The model was configured to Lake Onego's bathymetry. The TKE turbulent closure scheme (GGL90 package) (Gaspar et al.,





1990) was used to parameterize the sub-grid vertical mixing processes. The horizontal turbulent diffusion coefficients were
set to be constant. Because Lake Onego is located in the subarctic zone, the SeaIce package included in the MITgcm model
complex was used to simulate ice cover. To emulate the conditions of a freshwater lake, the salinity of the ice was set to zero.
The PTtracer package was used to solve the tracer equations of advection–diffusion required for biogeochemical variables.

**2.2 Study area**

The fully coupled hydrophysical-biogeochemical model was run on a spherical grid with a horizontal step of 1.079′ in latitude
and 2.331′ in longitude, which was approximately 2 × 2 km at the latitude of Lake Onego (Fig. 2). The z-coordinate was used
with a uniform vertical step of 2 m from the surface to the bottom. Figure 2 presents also main limnic regions of the lake
covering open waters and the most significant bays.

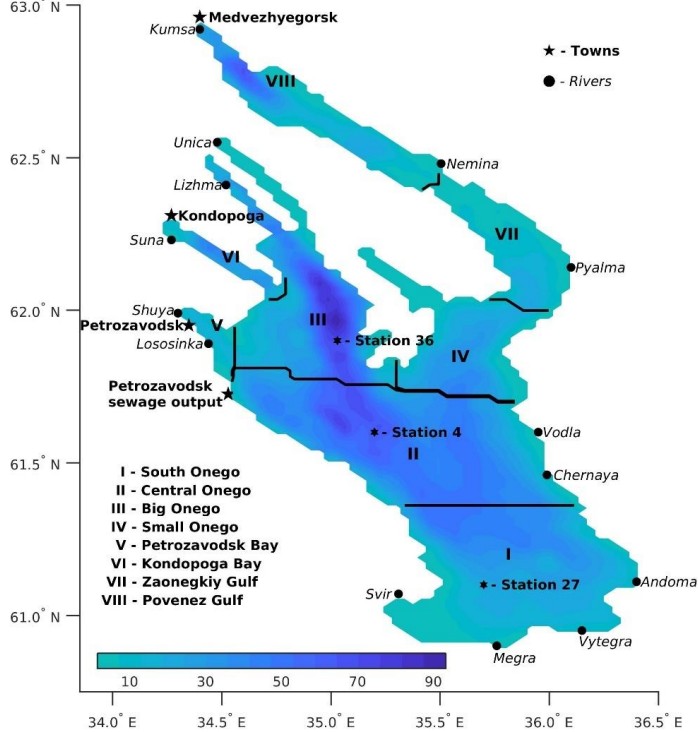

**Figure 2: Model bathymetry (meters), major limnic regions, rivers, settlements, industrial areas, and selected monitoring stations
are shown at the computational grid covering Lake Onego with a horizontal resolution of 2 × 2 km.**

**2.3 Initial and boundary conditions**

The initial conditions for the hindcast simulation of the biogeochemical dynamics of Lake Onego between 1985 and 2015
were generated during a 40-year spin-up simulation with the boundary conditions repeated for 1984, until a quasi-steady state

of seasonal dynamics, particularly for the sediment variables, was reached.

Atmospheric forcing was set based on the ERA-Interim reanalysis fields (https://www.ecmwf.int). The model used fields of
pressure, wind speed components, air temperature, humidity, short-wave and long-wave incoming radiation, and precipitation.





Monthly river runoff was reconstructed from a water budget based on field observations for a period of 60 years (Filatov, 2020). The total water discharge (Fig. 3a) was split among 13 rivers (see Fig. 2), with empirical coefficients of the contribution

of each river. The only river that drains Lake Onego into Lake Ladoga is the Svir River; its discharge was used to balance the reconstructed total river water input.

In the absence of reliable time-series of nutrient inputs entering Lake Onego from external sources (river runoff, atmospheric deposition, anthropogenic sources, etc.), reconstruction was carried out from a compilation of available published estimates. According to such estimates for 1965–2008, the rivers, wastewaters, and atmosphere contributed 66%, 23%, and 11% to total

phosphorus (TP) input, respectively, and 67%, 5%, and 28% to total nitrogen (TN) input, respectively (Sabylina et al., 2010). Sabylina (2016) compiled available information on the total content and inorganic fractions of nutrients in river waters and, assuming the similarity of concentrations in rivers that drain catchments with similar landscapes, provided reasoning for the aggregation of many streams and rivers into larger units flowing into Lake Onego. Correspondingly, we multiplied these concentrations by the monthly water discharge for the indicated 13 rivers, filling the gaps by linear interpolation between the

available discharge values. Dissolved organic content was further separated into labile and refractory components, assuming a 30% bioavailability of dissolved organic nitrogen (DON) and 90% bioavailability of dissolved organic phosphorus (DOP) (Stepanauskas et al., 2002; Isaev et al., 2020).

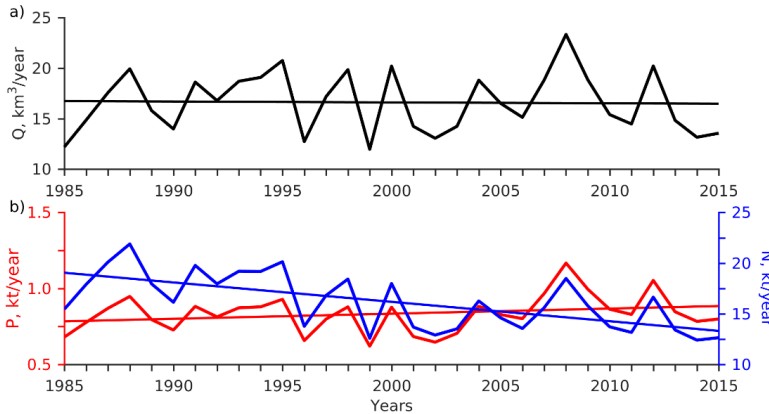

**Figure 3: Estimated annual river runoff (a) and full external inputs of total nitrogen and phosphorus (b).**

The main sources of direct anthropogenic pressure on Lake Onego are the three industrial areas associated with the coastal towns of Petrozavodsk, Kondopoga, and Medvezhyegorsk. In contrast to the other two, discharging directly into the top of the bay, the sewage output from Petrozavodsk industrial hub is located on the open coast south of Petrozavodsk Bay (see Fig. 2). According to estimates from various publications (Sabylina et al., 2010, Lozovik, 2016), the total annual input from these areas varies from 170 to 250 tons of TP and was approximately 2,650 tons of TN per year. Annual nutrient deposition from the

atmosphere on the lake surface ranges from 65 to 80 tons of TP and was approximately 2,260 tons of TN annually (Sabylina et al., 2010, Lozovik, 2016). Since the mid-2000s, commercial fish farming has been actively developing in the waters of Lake Onego, generating annual nutrient loads of approximately 40 tons and 228 tons of TP and TN, respectively (Lozovik, 2016). These fish farm inputs are comparable to those from the atmosphere and have been accounted for in boundary conditions since 2005.

Despite the almost trendless water discharge (Fig. 3a), the reconstructed external TP input showed an upward tendency, while the TN input distinctly decreased by approximately one-third (Fig. 3b). As a result, the bioavailable fraction of total nutrient





inputs, comprising inorganic and labile dissolved organic compounds, reached approximately 8,300 tons N/yr and 800 tons P/yr in the 2010s, thus, decreasing the bioavailable weight N:P ratio from 13.7 in 1985/89 to 10.4 in 2011/15.

## 3 Results and Discussion

The field data availability (its coverage, regularity, and frequency) for Lake Onego is not sufficient to allow extensive model–data comparisons similar to the ones made, for instance, for the Baltic Sea (Savchuk et al., 2012; Gustafsson et al., 2017; Isaev et al., 2020) or Great American lakes (Scavia et al., 2016 and references therein). Therefore, omitting the formal validation that usually precedes further modeling analysis, we start below with the results of the simulation, on the way involving in the analysis as much as possible and whatever data are available for Lake Onego. In addition to this highly insufficient information, 155 we also involved some relevant material from other boreal and even temperate lakes, situated in the differing hydrometeorological conditions and impacted by the differing land cover patterns and land use practices at their watersheds. Because of these wide differences, the order-of-magnitude comparability between modeled values and these arbitrarily compiled estimates should be considered as an indicator of plausibility of the simulation rather than its strict validation.

### 3.1 Long-term dynamics

#### 3.1.1 Important hydrophysical characteristics

The implementation of MITgem model for simulation of transport processes in the framework of SPBEM ecosystem model was justified by its successful application to large lakes such as Ladoga (Isaev and Savchuk, 2020), Michigan (Pilcher et al., 2015, Gloege et al., 2020), and Superior (Bennington et al., 2010). In this paper, aimed at the long-term and seasonal biogeochemical dynamics, we omitted analysis of the most hydrophysical characteristics and present model-data comparison 165 only for the ice coverage and surface water temperature as important integral indicators of the hydro-thermodynamics that significantly affect the ecosystem dynamics.

The realistic simulation of the ice cover dynamics, especially of its melting phase is a prerequisite for a timely reproduction of the phytoplankton spring bloom commencement, which in the model is determined by the light availability increasing due to the ice melt (see Fig. 9 in Sect. 3.2). A comparison of simulated dynamics to estimates obtained within visual–instrumental 170 approach from satellite data by a semi-supervised algorithm (Filatov et al., 2019) shows that the model captures both the timing and duration of seasonal ice phases rather accurately (Fig. 4). Most closely coincide the onset of freezing, observed from mid-December to early January, and the final ice clearing occurring from mid to late May.

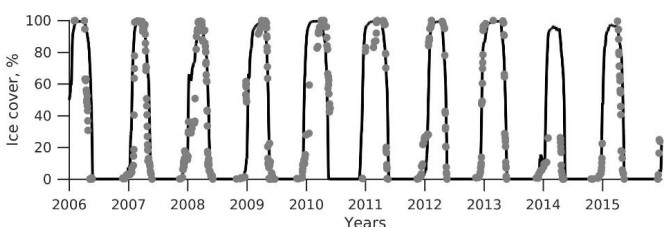

**Figure 4: Lake Onego ice coverage (% of the lake's surface) dynamics simulated (curve) and estimated from satellite data (dots).**


In addition to its significance in Lake Onego's hydrophysics, water temperature is a crucial factor in ecosystem dynamics, including its effects on the phenology and intensity of biogeochemical fluxes. Available water temperature measurements in Lake Onego were scarce and irregular. The most frequent observations were made in the summer months and occasionally in
May and September. For a model–data comparison, all available water temperature observations made at the surface during
calendar summers (June–August) of 1992–2007 (532 measurements) were pooled together regardless of the sampling location.
The model results for the same months and years comprised the daily average water temperatures from all of the computational
grid nodes. Over 70% of all measurements were collected from the coastal areas and bays; thus, the contribution of the
expansive and colder open waters to the observational statistics was lower than their contribution to statistics calculated from
simulation. Taking this expected bias into consideration, the simulated water temperature (11.85 ± 3.92 °C; median 13.15 °C)
matches the observations (13.05 ± 4.82 °C, median 14.40 °C) rather well. Model validation for the underrepresented open
areas was made for three stations (cf. Fig. 2), covered with regular observations from 1985 to 1989, while later only episodic
measurements were made. The best model-data comparability was found during the periods of heating and cooling of the lake,
while the larger discrepancies occur during summer maxima (Fig. 5). Apparently, with a grid cell height of 2 m and daily
averaging we cannot expect precise reproduction of the diurnal extremes in the surface temperature measured in the upper 0.5
m water layer. Such causes of the discrepancy could also partly explain the summer bias in the total model-data comparison
above.

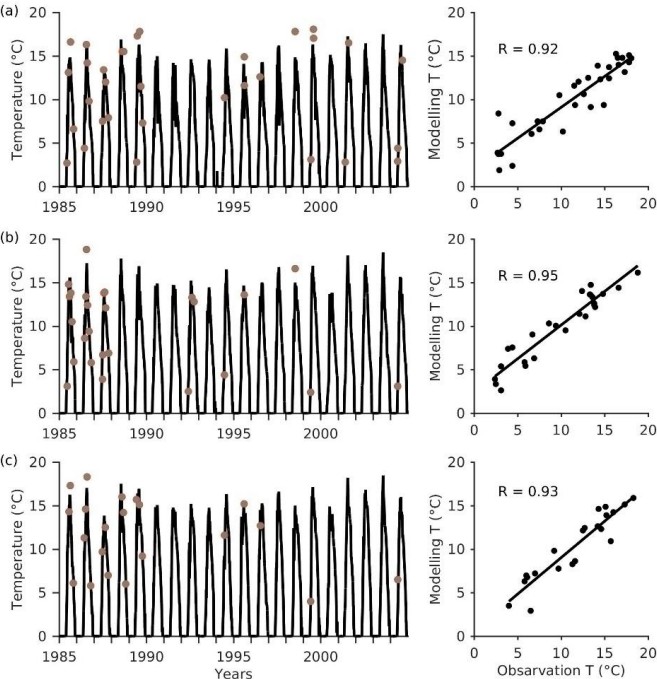

**Figure 5: Simulated (curve) and observed (dots) long-term dynamics of the surface water temperature at monitoring stations 36 (a),**
**4 (b), and 27 (c). Location of stations is shown in Fig. 2. Corresponding model-data linear regressions are shown to the right.**

The concept of "biological summer" can be characterized by its duration (BSD) and average water temperature (BST) and is
often used in hydrobiological studies. At Lake Onego, the day of the transition of the surface water layer temperature to over
a threshold of 10 °C was set as the phenological indicator of both the onset and end of "biological summer" (Rukhovets and
Filatov, 2010; Tekanova and Syarki, 2015). In the simulation, a long-term tendency of increasing BST was clearly seen through
significant interannual variations (Fig. 6). This tendency agrees well with both simulations and observations in the European
boreal zone (e.g., Ryabchenko et al., 2016; Kahru et al., 2016; Shiryaeva et al., 2018; Filatov, 2020).



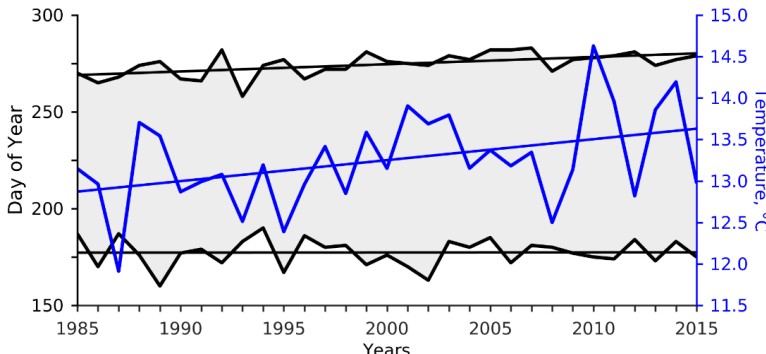

**Figure 6: Simulated duration (shaded area delimited by the start and end day of Gregorian Year) of the period of "biological summer" and its average surface water temperature with estimated trends (lines)**

The BSD in Lake Onego calculated from the simulation was 92 ± 9 days, within the range of 71–107 days. Based on sparse field data, BSD in the shallow areas (100–110 days) was markedly longer than that in open deep water (85–90 days) (Efremova et al., 2016). The prolongation of "biological summer" occurred chiefly because of its extension into autumn, from September 27 (mean date for the first 5-year period) to October 5 (mean date for the last 5-year period). A similar asymmetry between trends of spring warming and autumn cooling has also been found in the Baltic Sea (Kahru et al., 2016) and Karelian lakes (Efremova et al., 2016).

### 3.1.2 Ecosystem variables

Daily long-term dynamics of phytoplankton primary production and ecosystem variables averaged over the entire Lake Onego are presented in Fig. 7. Assuming the classical Redfield molar ratio of C:N:P = 106:16:1 being valid for the freshwater phytoplankton, including boreal oligotrophic lakes (Sterner and Elser, 2002; Reynolds, 2006; Sterner et al., 2008; Sterner, 2011; Bergström et al., 2018), the net primary production of diatom and non-diatom phytoplankton simulated in nitrogen weight units were converted into carbon weight units with a factor of 5.7 mg C mg$^{-1}$ N. Presented concentrations of inorganic and total nutrients were averaged over the whole water body, from the surface to the bottom. Consequently, the dissolved inorganic phosphorus (DIP), comprising summer phosphorus accumulation in the hypolimnion, was never fully depleted. The lake-wide averaged winter maximum values (Fig. 7 c, e) multiplied by the Lake Onego model volume (297 km$^3$) could be used to conveniently estimate total nutrient stocks. Diatom and non-diatom phytoplankton biomass, simulated as nitrogen units, were recalculated into wet weight units assuming a nitrogen content of 0.5% and 1% of the biomass of diatoms and non-diatoms, respectively (Menden-Deuer and Lessard, 2000; Reynolds, 2006). For zooplankton biomass, a nitrogen content of 1% was assumed (Raymont, 1984).

The simulated long-term dynamics steadily reproduce a distinctive seasonal pattern: the strong phytoplankton spring bloom transitions into the summer quasi-steady-state phase followed by minor autumn blooming. The short-term interannual variations were more pronounced in recent years in terms of primary production and biotic variables with higher plankton biomass (Fig. 7 a, b, d). The abiotic variables showed some apparent tendencies rather than distinctive periods (Fig. 7 c, e, f). The average concentrations of TP (7–13 mg P m$^{-3}$) and DIP (2–7 mg P m$^{-3}$) classify Lake Onego as bordering the oligotrophic and mesotrophic states, according to some classifications (Filatov, 2010; Dove and Chapra, 2015). Considerable decreases in TN and dissolved inorganic nitrogen (DIN) concentrations were mainly related to the reduction of external inputs (Fig. 3 b). Such a rapid response to the changing external inputs can be explained by the short nutrient residence times for the bioavailable





fractions that can be estimated for the water body from winter maxima and external inputs, approximately 8 and 2 years for nitrogen and phosphorus, respectively.

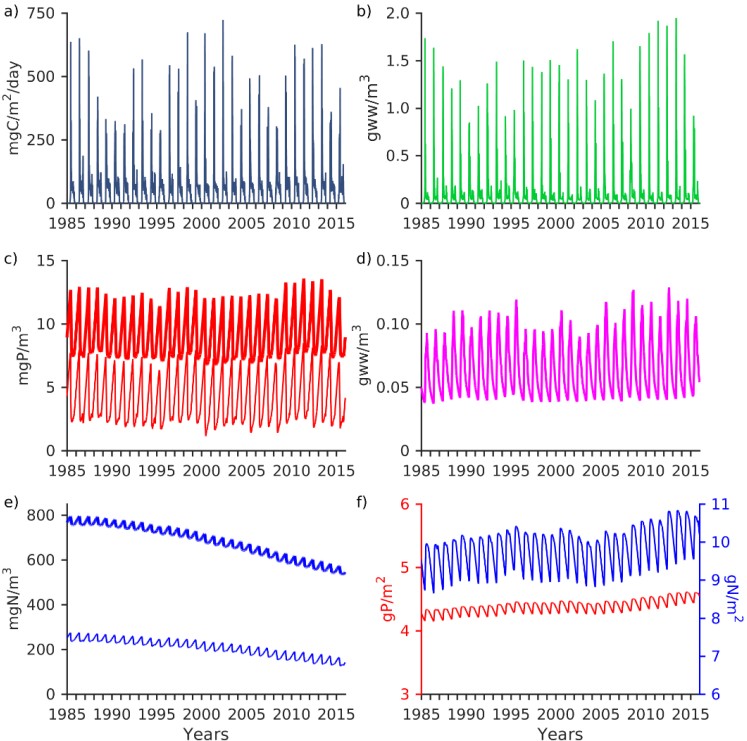

**Figure 7: Simulated long-term dynamics of (a) primary production (mg C m⁻² d⁻¹); (b) phytoplankton biomass (g wet weight m⁻³);** (c) phosphorus (mg P m⁻³) components: total phosphorous (TP; thick curve) and dissolved inorganic phosphorous (DIP; thin curve); (d) zooplankton biomass (g wet weight m⁻³); e) nitrogen (mg N m⁻³) components: total nitrogen (TN; thick curve) and dissolved inorganic nitrogen (DIN; thin curve); and (f) benthic nitrogen and phosphorus (g N (P) m⁻²).

Annual integrals of the simulated phytoplankton primary production of 17.0–20.6 g C m⁻² yr⁻¹ were almost invariable and matched the global mean estimate of 20 g C m⁻² yr⁻¹ for the large lakes situated to the north of 60° N (Alin and Johnson, 2007) but constituted only one-fifth of 94 g C m⁻² yr⁻¹ that was estimated for Lake Superior (Sterner, 2010). A small rise in plankton biomass (cf. Fig. 7, b, d) and annual production from the start toward the end of the simulation (from 18.5 to 20.2 g C m⁻² yr⁻¹) could be explained by the increased spring production, integrated over the time interval from the beginning of the year until the onset of biological summer, from 10.0 to 12.0 g C m⁻² yr⁻¹. In turn, this biotic growth occurred because of increased surface winter DIP accumulation (maximum values increased from 5.9 to 6.5 mg P m⁻³), which was caused by a combination of increased external loads (cf. Fig. 3b) and internal total P recycling. This comprised pelagic regeneration due to zooplankton excretion and organic phosphorus remineralization, as well as phosphate release from the sediments, from 457 to 492 mg P m⁻² yr⁻¹. A 1-week prolongation of the "biological summer" (cf. Fig. 4) insignificantly affected the small summer PP increase from 6.4 to 6.7 g C m⁻² yr⁻¹.

### 3.1.3 Spatial inhomogeneity

In contrast to the nearly invariable average seasonal dynamics in Fig. 7, the spatial distributions of variables and fluxes, which were largely determined by their proximity to external nutrient sources, were highly inhomogeneous (Fig. 8, note the



logarithmic scale). In the zones of elevated winter DIP concentrations (Fig. 8 b), annual primary production rates exceeding 25 g C m$^{-2}$ yr$^{-1}$ put large areas on the border of the oligotrophic status, while eutrophic areas were observed near the Petrozavodsk and Kondopoga industrial centers as well as along the coast south of Petrozavodsk Bay, where PP increases up

to 120–150 g C m$^{-2}$ yr$^{-1}$ (Farley, 2012; Filatov and Rukhovets, 2012; Efremova et al., 2019). The simulated inhomogeneous distribution matched the estimates obtained from the direct measurements; thus, the measured summer average daily PP decreased from 413 mg C m$^{-2}$ day$^{-1}$ at the top of Petrozavodsk Bay to 122 mg C m$^{-2}$ day$^{-1}$ in the open areas of the Bay. In the model, the PP for this entire limnic area was 265 mg C m$^{-2}$ day$^{-1}$. A similar decrease from 286 to 217 mg C m$^{-2}$ day$^{-1}$ observed in Kondopoga Bay matched the simulated PP of 244 mg C m$^{-2}$ day$^{-1}$. The average summer (June-September) daily PP of 111

and 72 mg C m$^{-2}$ day$^{-1}$, simulated for the central and southern Onego, respectively, corresponded well to measured levels of approximately 90 mg C m$^{-2}$ day$^{-1}$. In general, these offshore values were expectedly lower than the lake-wide mean primary production rates estimated for more eutrophic lakes Huron, Michigan, and Superior for 2010–2013 (216, 259, and 228 mg C m$^{-2}$ day$^{-1}$, respectively); although primary production in all depth zones (shallow, mid, and deep) were similar across the lakes (Fahnenstiel et al., 2016).

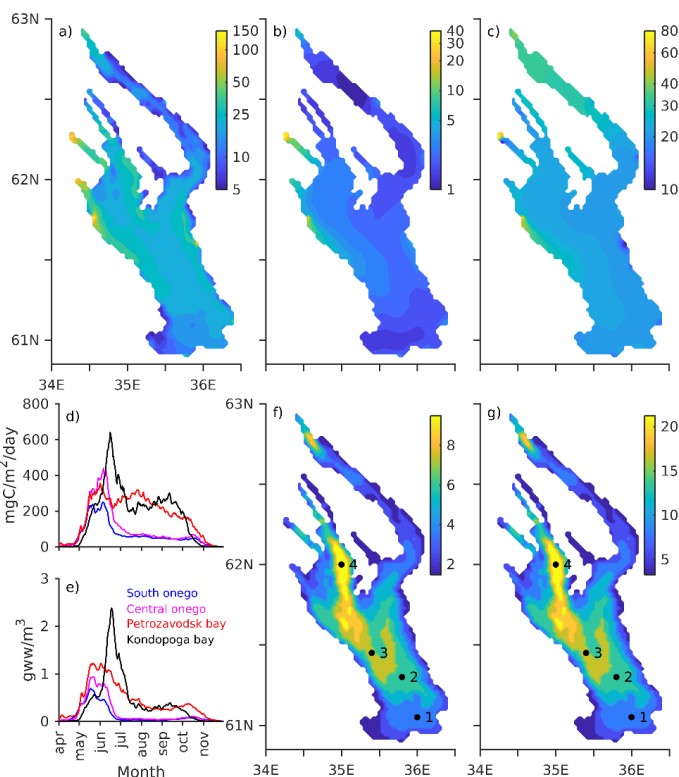


**Figure 8: Simulated long-term (1985–2015) averages of (a) annual primary production (g C m$^{-2}$ yr$^{-1}$); April surface concentration of (b) dissolved inorganic phosphorous (DIP; mg P m$^{-3}$) and (c) dissolved inorganic nitrogen (DIN; mg N m$^{-3}$) (note the logarithmic scale); average seasonal dynamics of (d) annual primary production (g C m$^{-2}$ yr$^{-1}$) and (e) phytoplankton biomass (g ww m$^{-3}$) in the different limnic regions of Lake Onego; and averages of (f) benthic phosphorus (BEP; mg P m$^{-2}$) and (g) benthic nitrogen (BEN; mg**

**N m$^{-2}$) with location of sites chosen for comparison with observations.**

The general levels and seasonal patterns of PP and phytoplankton biomass in the bays and open waters also differed (Fig. 8 d, e). A pronounced spring maximum in the oligotrophic central and southern Onego was followed by a weak autumn blooming





after a long summer minimum. In contrast, there were two equal peaks of primary production in Petrozavodsk Bay at the end of May and July. Similar seasonal dynamics were simulated in Kondopoga Bay, although both the spring and autumn peaks were somewhat delayed. Except for the simulated spring initiation of the vegetation season, these spatial and temporal developments were in good agreement with the phenology of the Lake Onego ecosystem deduced from field observations (Filatov, 2010; Rukhovets and Filatov, 2010; Tekanova and Syarki 2015; Efremova et al., 2019; Suarez et al., 2019).

The spatial distribution of primary productivity was well reflected in the sediments (Fig. 8 f, g). In the model, the sediment variables of benthic phosphorus (BEP) and nitrogen (BEN) were described according to a vertically integrated dynamic approach in areal units of mg P (N) m$^{-2}$ for the biogeochemically active surface layer of unspecified thickness (Savchuk and Wulff, 1996, 2001; Savchuk, 2002; Isaev et al. 2020). In geochemistry, nutrient content is usually calculated as a percentage ratio of the mass of phosphorus and nitrogen contained in a gram of ash-free dry sediment (Kalinkina and Belkina, 2010; Rukhovets and Filatov, 2018). For model–data comparison (Table 1), we chose four representative sites (see Fig. 8 g) and converted available measurements characterizing surface sediment layer of 5 cm thickness (Filatov, 2010; Kalinkina and Belkina, 2018) into model units, using the bulk density values obtained from similar seascapes (Carman et al. 1996). Taking into account all the uncertainties of such comparisons, starting with measurements being made over years from the mosaic distribution of real sediment types, and ending up with simplified and spatially invariable sediment parameterizations disregarding sediment types, the simulated areal concentrations could be considered plausible.

Table 1. Sediment characteristics and nutrient content measured at the sampling sites (see Fig. 8 g) chosen for comparison with the simulated benthic phosphorous (BEP) and benthic nitrogen (BEN)

| Site N | 1 | 2 | 3 | 4 |
|---|---|---|---|---|
| Depth, m | 26 | 48 | 54 | 86 |
| Sediment type | Muddy sand | Aleuritic mud | Mud | Mud |
| Bulk density, g cm$^{-3}$ | 1.5 | 1.2 | 1.1 | 1.1 |
| P content, % | 0.03–0.07* | 0.10–0.13* | 0.08–0.27 | 0.10–0.15 |
| P content**, g P m$^{-2}$ | 2.5–5.0 | 5.9–7.9 | 4.4–14.9 | 5.5–8.3 |
| Model***, g P m$^{-2}$ | 3.6 ± 0.1 | 6.1 ± 0.1 | 7.3 ± 0.2 | 9.8 ± 0.2 |
| N content, % | 0.05–0.15 | 0.2–0.3 | 0.2–0.6 | 0.3–0.7 |
| N content**, g N m$^{-2}$ | 3.8–11.3 | 12.0–18.0 | 11.0–33.0 | 16.6–38.6 |
| Model***, g N m$^{-2}$ | 8.2 ± 0.4 | 13.6 ± 0.7 | 16.3 ± 0.8 | 21.8 ± 1.1 |

* – assuming the bioavailable fraction constitutes 66% of the total phosphorus content (Table 1 in Kalinkina and Belkina, 2018)

** – calculated for a 5 cm-thick sediment layer

*** – mean ± s.d. calculated for 1985-2015

According to the simulation, the total stocks of N and P in the surface sediments of Lake Onego, which cover a bottom area of 9,266 km$^2$ and are actively involved in the contemporary biogeochemical cycling of the lake, amount up to 80,000 and 40,000 tons, respectively. The low weight N:P ratio of two can be attributed to the low nitrogen content (ca. 0.45% N of dry mass) in the contemporary deep-water sediments of oligotrophic lake, which, on the other hand, are highly enriched with phosphorus (ca. 0.25% P of dry mass) associated with iron–humic complexes (Kalinkina and Belkina, 2018; Kalinkina et al., 2020). These values are close to those of the contemporary Lake Winnipeg's sediment nutrient content of 0.4% N (Bunting et al., 2016) and 0.12%–0.26% P (Nürnberg and LaZerte, 2016). Regarding such integral characteristics and considering statements of the occurrence of many-fold increases in P sediment content during recent decades (Rukovets and Filatov, 2010),





one should be concerned as to where such increases occurred and which areas were enveloped. Less than 1,000 tons of annual TP load (cf. Fig. 3) would not be sufficient for any substantial alteration of the present-day TP integral sediment stocks.

### 3.2 Seasonal dynamics

Here, we begin with presenting long-term average seasonal dynamics (cf. Fig. 5) before analyzing its mechanisms. As there is no available data on the phenological phase of the spring phytoplankton bloom in Lake Onego, for the first time, we present this biogeochemically important phenomenon in more detail (Table 2).

Table 2. Simulated spring maxima of winter nutrient surface accumulation (mg P (N) m$^{-3}$), phytoplankton primary production (PP, mg C m$^{-2}$ d$^{-1}$), and biomass (Phyto, g wet weight m$^{-3}$) accompanied by the Gregorian day of year of its commencement
(mean ± SD) with the range of its occurrence.

|  | DIP | TP | DIN | TN | PP | Phyto |
|---|---|---|---|---|---|---|
| Maximum | 6.0 ± 0.4 | 11.4 ± 0.6 | 230 ± 34 | 710 ± 74 | 500 ± 128 | 1.4 ± 0.3 |
| Min/max | 5.0/6.8 | 10.2/12.6 | 164/279 | 574/808 | 288/723 | 0.8/1.9 |
| Day of year | 118 ± 12 | 124 ± 12 | 116 ± 11 | 121 ± 11 | 149 ± 12 | 139 ± 9 |
| Date range | Mar 27–May 15 | Apr 2–May 21 | Mar 28–May 14 | Mar 31–May 19 | May 3–Jun 13 | May 3–Jun 5 |

The maximum concentration of inorganic nutrients is reached when the winter accumulation due to pelagic and sediment remineralization is interrupted because of increasing nutrient uptake by developing phytoplankton. Thus, the time of maximum inorganic nutrients can be considered the onset of spring blooming, which, in the model, begins at the end of April with half-
month interannual fluctuations (Table 2). Within these fluctuations, there is a tendency for an earlier spring onset, from the beginning of May to the end of April (cf. Fig. 6). For phytoplankton primary production, dominated in spring by diatoms, it takes an average of 1 month to reach its maximum, commencing over the years as early as May 3 and as late as June 13 in some years, without any long-term trend. Contrary to expectations, based mainly on our marine modeling experience, the maximum phytoplankton biomass averaged over the entire lake was attained statistically earlier (up to a month earlier in some
cases) than that of the PP maximum.

Although a spring bloom phase is typical in boreal and arctic lake phenology (e.g., Vehmaa and Salonen, 2009; Hampton et al., 2015, 2017; Maier et al., 2019; Yang et al., 2020), there is almost no information on daily spring PP values measured in limnological conditions, similar to that of the boreal oligotrophic Lake Onego, situated between 61° N and 63° N. In Lake Baikal, areal primary production was measured as 800 ± 310 mg C m$^{-2}$ d$^{-1}$ under exceptionally transparent ice (Shchur and
Bondarenko, 2012). In Lake Superior, a PP rate of 220 mg C m$^{-2}$ d$^{-1}$ was measured on April 30, 2008 (Sterner, 2011), while rates of 150 mg C m$^{-2}$ d$^{-1}$ and 200 mg C m$^{-2}$ d$^{-1}$ were reported from ice-free April 1999 and partly icy June in 2000 (Urban et al., 2005). These measurements are extremely similar to the spring maxima simulated with a medium-complexity ecosystem model for 1997–2001, reaching up to 200 mg C m$^{-2}$ d$^{-1}$ (Bennington et al., 2012). As can only be expected, Lake Onego's spring maximum PP rates are approximately twice of the rates in Lake Superior, where median TP and DIP concentrations
were less than 3 mg P m$^{-3}$ and 0.7 mg P m$^{-3}$ (Urban, 2009), that is, less than a third of Lake Onego's winter maximum stocks, providing for a much higher autochthonous production of organic matter (cf. Fig. 7, Table 2). Higher spring PP rates of up to 1200 mg C m$^{-2}$ d$^{-1}$ were measured in 1985–1987 in three small Northern Wisconsin lakes located at 46° N and having comparable to Lake Onego spring TP concentrations (Adams et al., 1993). Similarly high spring rates of 800–1200 mg C m$^{-2}$ d$^{-1}$ were measured in the offshore region of southeastern Lake Michigan in the 1980s and the 1990s (Fahnenstiel et al., 2010).



After the spring bloom, the simulated ecosystem segued into the summer phase (see Fig. 8 d, e and Fig. 9 below), characterized by progressively decreasing phytoplankton biomass to levels of approximately 0.1 and 0.4 g ww m$^{-3}$ in the open waters and the major bays, respectively. Phytoplankton PP in these regions remained in a quasi-steady state, at less than 100 and 200–300 mg C m$^{-2}$ d$^{-1}$ in the open waters and major bays, respectively. During summer, the simulated lake-wide average diatom biomass of 0.3 g ww m$^{-3}$ still dominated the phytoplankton community, similar to biomass values of 0.01–0.7 g ww m$^{-3}$ reported from

the field observations (Tekanova and Syarki, 2015). The concurrently developing simulated non-diatom complex reached its maximum biomass of 0.05 g ww m$^{-3}$ in July–August, which is close to the reported maximum of 0.034 g ww m$^{-3}$. Organic matter produced by phytoplankton accumulated in the water column to levels of up to 5 mg P m$^{-3}$ (cf. Fig. 7 c). Its particulate fraction (detritus) exceeded 1–2 mg P m$^{-3}$ in the epilimnion and, together with phytoplankton, was grazed by zooplankton, whose biomass reached its maximum of 0.1–0.2 g ww m$^{-3}$ in July–August. The average value of zooplankton biomass

measured during the vegetation period was 0.1–0.33 g ww m$^{-3}$ (Rukhovets and Filatov, 2010). Unconsumed and non-mineralized detritus sediments on the lake bottom, where it is partly buried and mineralized generating characteristic seasonal dynamics (cf. Fig. 7 f). With a sediment weight C:N ratio of approximately 10 (Bunting et al., 2016; Kalinkina and Belkina, 2018), the amplitude of seasonal variations of 1 g N m$^{-2}$ corresponds to organic carbon seasonal variations of approximately 10 g C m$^{-2}$. Primary production during autumn diatom blooming and phytoplankton biomass was simulated to values of up to

130 mg C m$^{-2}$ d$^{-1}$ and 0.2 g ww m$^{-3}$, respectively, which constitutes triple that reported as 40 mg C m$^{-2}$ d$^{-1}$ and quarter that reported as 0.8 g ww m$^{-3}$ (Tekanova and Syarki, 2015).

Regarding phenological generalization (Syarki and Tekanova, 2008; Tekanova and Syarki, 2015), it is worth noting that a neglect of the spring phytoplankton bloom phenomenon as well as the normalization by the maximum value measurements available only from the post-bloom "summer window", which is rather typical in lake research (e.g., Hampton et al., 2015),

has resulted in a number of misleading implications. Here, we amend such implications with the following conclusions: a) the seasonal maximum of daily PP and the greatest contribution to annual PP occurred during spring, not summer; b) the initiation of the vegetational season was due to increasing light availability, with temperature dynamics playing a later role, due to thermal convection and augmented nutrient regeneration (see below); and c) the spring bloom was produced by diatoms, which contributed most also to summer PP in all regions except for the Kondopoga and Zaonegskiy bays, where their contributions

were approximately the same as that of non-diatoms.

The mechanisms of seasonal dynamics in boreal lake ecosystems are well known, although the under-ice and melting-ice phases have often been overlooked (Hampton et al., 2015). In the model developed in this study, seasonal ecosystem dynamics were determined by a close interaction between physical phenomena including transports, and biogeochemical processes (Fig. 9). Under the condition of reverse temperature stratification (Fig. 9 a), hampering the downward mixing of phytoplankton

cells, winter accumulation of phosphates in the water column, occurring mostly owing to sediment release of mineralization products (Fig. 9 b), was interrupted by a mounting phosphate uptake by developing phytoplankton. The increasing phytoplankton growth rate was determined by the rise in light availability occurring in the model with the ice melt. Phytoplankton PP and biomass quickly reached their maxima (Fig. 9 c, see also Table 2) and continued to develop with the support of phosphates brought upwards by total lake turnover when water temperature reached its ubiquitous value of 4 °C.

Later on, when the surface layer was depleted of DIP, the phytoplankton continued autotrophic activity in the deeper water layer under the thermocline (e.g., Fahnenstiel and Scavia, 1987; Sterner, 2010; Rowe et al. 2015), where light needed for photosynthesis was still sufficient. DIP also became available due to upward transport from the deeper layers, where DIP had been accumulating owing to organic matter mineralization in the water body and sediments. Organic matter produced by autotrophs and distributed through the water column as both live cells and detritus feed zooplankton, allowing them to reach



their maxima in August (Fig. 9, d). Vertical water convection associated with autumn cooling brought up enough phosphates
       to support a minor autumn bloom (Fig. 9, b, c).

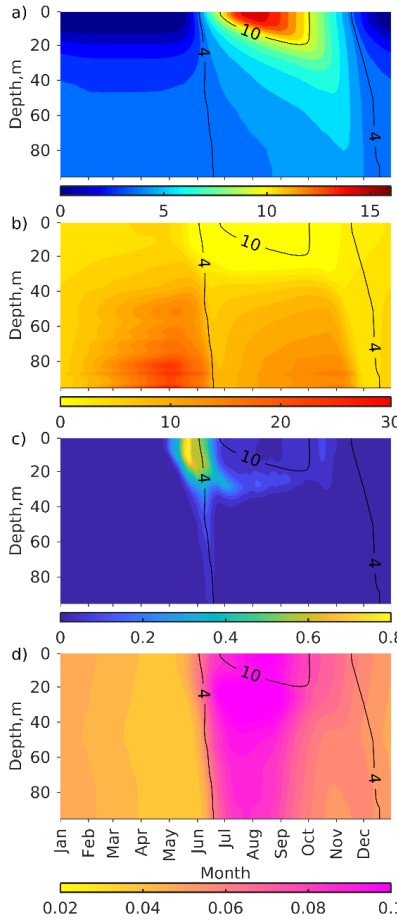

**Figure 9: Simulated long-term (1985–2015) basin-wide average seasonal time-depth dynamics of water temperature, °C (a); DIP, mg P m⁻³ (b); phytoplankton biomass, g ww m⁻³ (c); and zooplankton g ww m⁻³ (d). Contour plots are overlaid with isotherms of 4**

**°C (temperature of maximal water density) and 10 °C (delimiting "biological summer").**

As organic matter oxidized via limnetic and sediment biochemical mineralization as well as zooplankton catabolism consumed oxygen, the dynamics of its concentration in the hypolimnion (not shown) were reciprocal to the dynamics of phosphate as a product of oxidation (Fig. 9 b). In deep water layers, the winter maximum oxygen reserve, generated by the turnover of the water column in December, was being continuously depleted until the May minimum temperature. The oxygen reserve was

then replenished by the late June turnover and reached the second minimum in September; however, the oxygen consumption was slow; for instance, only 40 and 50 g $O_2$ m⁻² yr⁻¹ was taken up by the water and sediments, respectively, compared with 140–560 g $O_2$ m⁻² yr⁻¹ consumed by sediments in western Lake Erie (Boedecker et al., 2020). Seasonal amplitude was small, and simulated oxygen concentration in the deepest grid's depth of 90 m in Big Onego (site 4 in Fig. 8 f, g) alternated interannually between maximums of $11.98 \pm 0.22$ g $O_2$ m⁻³ and minimums of $7.00 \pm 1.84$ g $O_2$ m⁻³. These averages far exceeded

the oxygen deficit values that would trigger and maintain the "vicious circle" of eutrophication, which, in the Baltic Sea, is driven by the intensification of denitrification under hypoxic conditions and phosphate release from anoxic bottoms, thus,





lowering the DIN:DIP ratio and leading to the expansion of diazotrophic cyanobacteria blooms (Vahtera et al., 2007; Savchuk, 2018). The "vicious circle" incidence seems unlikely in Lake Onego, because of DIN concentrations being high and deep layers being ventilated twice a year (cf. Fig. 9 a). For example, explanations for the ongoing eutrophication and emergent
cyanobacteria blooms in large boreal lakes, such as Lake Winnipeg and Lake Superior, are sought in the immediate effects of changing external inputs and impacts rather than in the evolution of internal biogeochemical cycling (e.g., Schindler et al., 2012; Zhang and Rao, 2012; Bunting et al., 2016; Sterner et al., 2020; Howarth et al., 2021).

These described general interactions are largely occurring in the vast open deep-water areas. In coastal areas shallower than 30 m, which occupy approximately 53% of Lake Onego's area, especially in the bays, dynamics differ (cf. Fig. 8 d, e), mainly
because of the continuous support of PP by nutrients regenerated at, and released from, the bottom sediments situated within the epilimnion. However, such detailed spatial considerations are beyond the scope of this study.

**3.3 Biogeochemical fluxes and budgets**

There is almost no historical or contemporary information about external and, especially, internal biogeochemical fluxes based on field measurements, except for instances of compiled nutrient inputs and PP estimates. Instead, we present two subsets of
5-year averages obtained by the annual integration of the simulated three-dimensional fields of concentrations and selected biogeochemical fluxes over the entire Lake Onego (Table 3, cf. Fig. 1 and 2). We used these subsets, reconstructed by internally consistent numerical reanalysis, to demonstrate and highlight the most important features of the biogeochemical cycles of Lake Onego.

First, the comparison of external exchanges with internal cycling clearly showed that nutrient cycles were driven mostly by
internal biogeochemical processes. Annual integrals of both the external inputs of nutrients into Lake Onego and their removal: a) by a permanent sediment burial, b) with the Svir River outflow, and c) by denitrification, were several times smaller than the stocks already accumulated and cycling in the water body and in the bottom sediments. Only 17% of P and 25% of N were provided to autotrophs externally, while the rest was supported by inorganic nutrients regenerated in the water column and released from the sediments. The prevalence of internal cycling over external impacts is common in neighboring Baltic Sea
ecosystems (e.g., Savchuk, 2002, 2005, 2018) and in the remote Lake Superior (Urban, 2009).

The balance between external exchanges and internal cycling is usually expressed with a nutrient residence time, calculated as the ratio of integral nutrient stock to the amount causing the annual change of the stock, where total external input traditionally serves as a denominator. In addition, by estimating nutrient residence times from long-term reanalysis with seasonal and interannual variations (see Fig. 3 and 7), we could also relate the total stock to a half-sum of sources and sinks. For instance,
limnetic P residence time based on the 1985–2015 averages was calculated as the integral TP amount (2,681 t P) divided by the half-sum of total P input (834 t P yr$^{-1}$), export via the Svir River (131 t P yr$^{-1}$), sedimentation out of the water body (3,371 t P yr$^{-1}$), and phosphate release from the sediments back into the water body (2,401 t P yr$^{-1}$).

Limnetic residence times, calculated relative to either only external inputs or accounting for the water-bottom exchange, were short (Table 3), explaining the fast responsiveness of Lake Onego's waters to interannual variations of external effects (see
Fig. 3 and 6). The residence time for an external P input of 3.6 years has also been estimated on the basis of field observations (Lozovik 2016). Although with some controversy generated by the different methods of calculation, similar P residence times, from a few months to several years, have been estimated for Lake Superior (Urban, 2009). However, N residence time, despite being reevaluated by Urban (2009) to 55 years from an older estimate of 160 years, is still approximately ten times longer in Lake Superior than in Lake Onego. Such estimates, which are traditionally made only for water bodies and do neglect nutrient
stocks and processes in the surface biogeochemically active layer of the bottom sediments, can be quite misleading. For the





entire Lake Onego ecosystem, longer residence times of 47 and 17 years for P and N, respectively, were calculated when accounting for both the water and sediment stocks. Consequently, in contrast to short-term responses to external inputs, the long-term reaction of the entire phosphorus-limited Lake Onego ecosystem would be slow, as seen in Fig. 7.

Table 3. Simulated annual biogeochemical fluxes and total amounts of nutrients, integrated over the entire Lake Onego and
averaged for the start and end 5-year time intervals. Nutrient residence times are estimated for the entire 1985–2015 simulated time interval.

| Parameters | Phosphorus fluxes, t P yr⁻¹ | | Nitrogen fluxes, t N yr⁻¹ | |
|---|---|---|---|---|
| | 1985–1990 | 2011–2015 | 1985–1990 | 2011–2015 |
| External exchange | | | | |
| Total nutrient input | 814 | 862 | 18,662 | 13,649 |
| Dissolved inorganic | 395 | 431 | 5,724 | 4,950 |
| Labile dissolved organic | 356 | 366 | 4,573 | 3,338 |
| Export via Svir River | 136 | 149 | 9,645 | 5,605 |
| Permanent burial | 870 | 917 | 1,942 | 2,073 |
| Denitrification | | | 7,221 | 7,638 |
| Internal cycling and bottom-water exchange | | | | |
| Uptake by autotrophs | 4,327 | 4,708 | 30,289 | 32,958 |
| Total limnetic recycling | 1,586 | 1,731 | 13,628 | 13,189 |
| Sedimentation | 3,278 | 3,542 | 23,048 | 24,920 |
| Sediment release | 2,334 | 2,483 | 13,669 | 14,897 |
| Total amount*, tonnes | | | | |
| Water column | 2,641 | 2,832 | 224,711 | 168,598 |
| Sediments | 39,743 | 41,757 | 88,798 | 94,190 |
| System | 42,383 | 44,588 | 313,509 | 262,788 |
| Nutrient residence time (1985–2015), years | | | | |
| Limnetic** | 0.8 (3.2) | | 6.4 (12.4) | |
| Sediments | 12.2 | | 3.8 | |
| System | 46.9 | | 17.4 | |

\* - Average calculated on January 1 of every simulated year.

\*\* - Average total amount divided by the half-sum of sources and sinks (average total amount divided by average external input)

The lower buffer capacity with respect to nitrogen was explained by denitrification, which is inevitably set at some depth in the pore waters over the entire sediment area but is sometimes overlooked in studies of large boreal lakes (e.g., Urban, 2009; Finlay et al., 2007; Scavia et al., 2014; Bunting et al., 2016). Nowadays, denitrification in Lake Onego removes 56% of the total nitrogen input or over 90% of the bioavailable nitrogen input. The Lake Onego long-term lake-wide mean denitrification rate of 0.8 g N m⁻² yr⁻¹ belongs within the range 0.12–7 g N m⁻² yr⁻¹ that was compiled for oligotrophic and oligo-mesotrophic
lakes by Saunders and Kalff (2001). These authors also reported an average rate of 13 g N m⁻² yr⁻¹ when measured at multiple littoral sites in the oligotrophic Lake Memphremagog (located at approximately 45° N) with water temperatures exceeding 18 °C. Similarly high rates (13 g N m⁻² yr⁻¹ in 2016 and 5 g N m⁻² yr⁻¹ in 2017) were measured in the eutrophic western basin of Lake Erie (Boedecker et al., 2020). Measurements at 86 different stations across lakes Superior, Huron, Erie, and Ontario varied both spatially and temporally from 0.01 to about 400 g N m⁻² yr⁻¹, covering wide ranges within each lake and exhibiting





significant overlapping; for example, the denitrification rates measured in the Lake Superior bays were closer to the rates in Lake Erie and Huron than at other deep sites in Lake Superior (Small et al., 2016). Evidently, spatial inhomogeneity and seasonal variations also exist in Lake Onego (see Figs. 7 f & 8 g). For example, at the deepest site 4 (see Fig. 8 g) the denitrification rate reached up to $5 \pm 0.3$ g N m$^{-2}$ yr$^{-1}$ (mean ± SD) in July–August, i.e. after sedimentation of the freshly produced detritus.

Together with burial, denitrification removes approximately 40% of organic nitrogen that reaches the bottom surface (cf. Table 3); the rest is returned to the water column at an average rate of 1.5 g N m$^{-2}$ yr$^{-1}$, which is comparable to 0.9 g N m$^{-2}$ yr$^{-1}$ calculated here as a difference between nitrification and denitrification rates measured in the sediment core incubation at Lake Superior (Small et al. 2014). In Lake Onego, an almost equal share of inorganic nitrogen was provided by the total recycling of 1.4 g N m$^{-2}$ yr$^{-1}$ in the water body.

In Lake Onego, the bottom release of phosphorus (0.26 g P m$^{-2}$ yr$^{-1}$), which returns approximately 70% of the annual sedimentation to the water column, is more important for internal cycling than limnetic phosphorus recycling (0.18 g P m$^{-2}$ yr$^{-1}$). These values are within the same order of magnitude as mineralization rates of organic phosphorus (0.4–1.2 g P m$^{-2}$ yr$^{-1}$) in the water column of Lake Superior (Urban, 2009) and bottom release rates of 0.6 g P m$^{-2}$ yr$^{-1}$ in western Lake Erie (Boedeker et al., 2020), or extrapolations of 0.4 g P m$^{-2}$ yr$^{-1}$ which were estimated over Lake Winnipeg by Nürnberg and LaZerte (2016).
In the neighboring oligotrophic Lake Ladoga, an extensive sediment survey resulted in a range of 0.02–0.24 g P m$^{-2}$ d$^{-1}$ (Ignatieva, 1996).

**4 Conclusions**

1. Despite a long history of extensive research, the drivers and components of Lake Onego's biogeochemical cycles are still insufficiently studied. Especially damaging such a deficit of knowledge could be to our capability of forecasting possible
changes in the lake ecosystems in response to natural variations and anthropogenic impacts. To enable a quantitative description of the past, present, and, eventually, the future state of Lake Onego, we adapted and successfully implemented a three-dimensional biogeochemical model, considering the obtained results as a form of reanalysis.

2. The model was used to reconstruct three decades of Lake Onego ecosystem dynamics with daily resolution. Although the paucity of observations did not allow either formal model validation or the statistical demonstration of its skills, the comparison
to all available information from Lake Onego and to a range of published estimates for other large boreal lakes led us to believe in plausibility of simulation.

3. This reanalysis generated regionally new phenological knowledge that was previously missed due to a lack of regular winter observations. The analysis of the simulation quantified that the spring phytoplankton bloom, which was previously overlooked, occurred at the beginning of May and contributed to approximately half of the annual primary production of the lake. This
bloom was triggered by increasing sun radiation rather than an insignificant rise in water temperature.

4. Coherent nutrient budgets built on the simulated stocks and biogeochemical fluxes integrated over the water body and (often neglected) bottom sediments revealed a high buffer capacity of Lake Onego, which is also reflected in long phosphorus and nitrogen residence times, 47 and 17 years, respectively. Effective buffering is defined by an internal biogeochemical cycling of nutrients within and between the pelagic and benthic subsystems, which is much more intensive than external inputs and
exchange.



5. The shorter nitrogen residence time was explained by sediment denitrification, which is often ignored in studies of other large lakes, together with insufficient attention to the sediments and their role in biogeochemical cycling.

6. The basin-wide biogeochemical reanalysis also eliminated a historical disparity in attention and, hence, in observations and knowledge, between bays (especially Petrozavodsk and Kondopoga bays) and vast open-water areas. Bays occupy relatively
small bottom areas and water volumes and, consequently, are responsible for only a small fraction of biogeochemical phenomena and fluxes.

**Author contribution**

All authors contributed substantially to the study's conception, read, and commented the final submitted manuscript. OPS participated in planning and analysis of simulation, wrote original draft and assimilated suggestions from co-authors. AVI
adapted the model, made computations, processed outputs, and analyzed results. NNF was overall project lead, provided available observations and analyzed results.

**Competing interests**

The authors declare that they have no conflict of interest.

**Acknowledgments**

We are grateful to N. M. Kalinkina and T. F. Tekanova for their helpful discussions on the purpose and results of this study. We thank A. F. Balaganskyi for the river water discharge data. The author O.S. was supported by the Swedish Agency for Marine and Water Management through their grant 1:11—Measures for the marine and water environment. The author A.I. conducted the present study within the framework of state assignment (Theme No. 0128-2021-0014). The author N.F. conducted the present study within the framework of state assignment (Theme No. 0185-2021-0007).

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
