# Peer review of "Modeling of the large-scale nutrient biogeochemical cycles in Lake Onego"

_Biogeosciences, 2021_

## Referee Comment (RC2)

**Review of: Modeling of the large-scale nutrient biogeochemical cycles in Lake Onego**

I previously reviewed this paper when it was submitted to Limnology & Oceanography. Of interest to me was to understand if the authors had responded to my earlier comments, the comments of another reviewer and the handling editor. These comments should have translated to changes throughout the paper. In the first instance, I noted the Abstracts of the two papers (L&O and Biogeosciences) were identical. I have gone through the reviewers' comments from the earlier review (in blue) and added new text (in black) that reflects whether I consider the earlier comments are adequately dealt with in the current submission to Biogeosciences.

Reviewer: 1
This paper presents a largely theoretical physical-biogeochemical simulation of Lake Onego for a period of 40 years. The paper seeks to use a modelling approach to collate some of the relatively sparse and disparate sources of information available on the lake. While this approach is commendable, it needs to be well supported with a sound underpinning modelling framework. Such a framework would involve:
- Being sure to collate the available and relevant sources of information available on the lake. This was not done adequately in my opinion and several of the papers that were part of a special issue on Lake Onego (Inland Waters Vol 9, Issue 2), and contained relevant information, were not cited, while at the same time the authors stated that "there is almost no empirical information on the major biogeochemical variables and fluxes".

The authors have partially addressed this point – noting the inclusion of the Efremova et al. 2019 paper (as cited by the authors).

- A sound modelling process involves calibration and validation against concentrations and/or biogeochemical fluxes. The comparison of temperature made by the authors: "Taking this expected bias into consideration, the simulated water temperature (11.85 ± 3.92 °C; median 13.15 °C) matches the observations (13.05 ± 4.82 °C, median 14.40 °C) well" is inadequate and direct comparisons (observations vs. simulation output) should have been made together with the relevant underpinning errors statistics (e.g., R2, RMSE, PBIAS, etc.).

The authors have not in my opinion adequately addressed this point. They suggest that "the simulated water temperature (11.85 ± 3.92 °C; median 13.15 °C) matches the observations (13.05 ± 4.82 °C, median 14.40 °C) rather well" but the reader is not told what the error statistic (±) relates to and it is not clear why model output was not aligned with comparable location of stations in claiming that there was bias due to 70% of measurements coming from coastal areas and bays. It looks from Fig. 5 like a comparison was done for three specific stations, but the R values are rather poor for temperature and a 1:1 line should also be shown to examine if there was systematic bias. I note that the authors have included a figure (Fig. 4) showing observed and simulated ice cover but give no statistics for goodness of fit and no detail about the way in which ice cover was simulated. Figure 6 caption should explain the black and blue lines and give the temperature for the 'biological summer'. Most importantly, the reader has no information if vertical thermal stratification is captured in the model; are there really no vertical profiles of temperature?

I appreciate that data may have been sparse but there are methods to counter this, e.g., use of remote sensing to provide optically active surface water constituents that can be compared with model output. Sensitivity analysis is also another useful approach to develop confidence in the

model simulations and be able to define a range of output as part of a model error analysis. Bootstrapping approaches are also useful for sparse data. Without this the model simulations become a largely theoretical exercise because we do not know the accuracy of the simulation output.

I didn't see a lot of additional effort to address suggestions around sensitivity analysis or the use of remote sensing data to demonstrate spatial variability in the model.

On line 215-216, it is stated that "Presented concentrations of inorganic and total nutrients were averaged over the whole water body, from the surface to the bottom". I understand why this might be done for a winter mixed period, but not for summer. I'm unclear on the following sentence as well, and it seems to be that distributing DIP through the water column in summer would lead to some large inaccuracies. Indeed, earlier it is stated (line 211) that "…variables averaged over the entire Lake Onego"; was this a volumetric average or was it a vertical average for the water column? The authors seem to add doubt between observations and model assumptions with the statement that: "Consequently [because the authors chose to average nutrient concentrations over the whole water body], the dissolved inorganic phosphorus (DIP), comprising summer phosphorus accumulation in the hypolimnion, was never fully depleted'.

- Almost no information is given on the parameters that go into the model. Parameters like sediment nutrient release rates, deoxygenation rates, algal growth rates, etc., need to be provided in any modelling exercise; they serve as a basis for future work and refinement (e.g., in experimental work) and they should generally fit within literature ranges. A hint of parameterization is given in the sediment N and P content values given in Table 1 but it is notable that the model sediment N content is mostly greater than the range given for the measured values – and this is mostly the case for sediment P also. This raises some major question marks about the N and P mass balances that are given for the lake.

Information on parameters is still not given. There is a great deal of uncertainty for the reader relating to the way in which the biogeochemical model was calibrated. The paragraph in the Ecosystem variables section (lines 211 to 222) did not alleviate these concerns.

- The reader is given no overview of measured forcing data inputs into the model. For example, the 'estimated' river runoff and N and P loads (Fig. 3) should have had the observations included also. Further, the meteorological variable inputs to the model should have been clearly specified. There were quite a few typographical errors through the paper (e.g., spelling mistakes in Fig. 1) that would need to be corrected in future iterations of this paper.

The authors now provide a brief description of inflow, outflow and meteorological forcing data for the model input.

I would suggest not mixing the results and discussion into a single section, which would help to add clarity.

Not done. This paper could easily be improved by separation of Results and Discussion sections.

 In other instances, elements of Introduction should not appear in the Methods. The Introduction would benefit from a clearly defined scientific hypothesis or test being put forward (in the last

paragraph of the Introduction), i.e., this would form the basis of the model testing or scenario generation.

No clear scientific test or hypothesis presented.  What was the purpose or objective of this paper?

What efforts were made to validate the discharge and nutrient concentrations measurements? It is not clear why the reviewer cannot see the interpolated load values (Fig. 3) plotted against the actual measurements (as points), so that the reader can see the frequency of measurements. Were the so-called "upward tendency" and "distinctly decreased" changes in loads with time actually significant? How were concentrations of dissolved nutrients in the inflows determined?

No formal calibration and validation of the model is carried out by the authors, raising doubts about the predictive capabilities of the model and more effort (e.g., remote sensing) should have gone into supporting this process.

It is not clear what the authors are pointing out in the following: "Taking into account all the uncertainties of such comparisons, starting with measurements being made over years from the mosaic distribution of real sediment types, and ending up with simplified and spatially invariable sediment parameterizations disregarding sediment types, the simulated areal concentrations could be considered plausible".  It is not clear what would be plausible or implausible.

---

## Author Comment (AC1)

**General comments**

In the paper, the authors use a 3D thermo-hydrological and biogeochemical model to simulated the nutrient cycles in Lake Onego.

They reconstruct 3 decades and made a lot of comments and conclusions on the simulated results.

The most important problem they have to face is that there are very few data available to validate their model. The authors are fully aware of this and justify their work and the use of the 3D model on this basis. The knowledge gained and integrated into the model should be able to compensate in some way for the lack of data. The authors go so far as to say that the hindcast results can be used as a form of re-analysis.

Thank you. We appreciate the thorough review and suggested changes, especially your clear understanding of the problems arising because of extreme paucity of biogeochemical data. Excuse us, please, for sometimes kind of didactic tone in reminding certain basics of mechanistic modeling, which we had to remind time and again in discussions even with some of our fellow modelers. To avoid incontinences with multiple attachments, we structure this single PDF in the following succession: 1) Full replays and explanations to your Review; 2) suggested new comparisons with measurements of vertical temperature distribution (new Fig.6) and primary production (new Table 1 and Fig. 10), as well as new Table A1 for the Appendix, presenting recalibrated phytoplankton constants; 3) something that we call Clarifying considerations (referred to as CC) that contain some material (maps, pictures, etc.) to which we refer to- but still do not intend copying it into the manuscript.

**According to me, there are several problems with the approach:**

1. calibration: models outputs are very sensitive to the parameters values which differs from one lake to the other. The authors have not performed any kind of calibration. They have used the parameters set calibrated on data of the Baltic Sea which is very different from the lake Onego. Adding to this the lack of validation data, the simulations used cannot be considered reliable.

We have both explanations and additions to the text related to calibration.

First of all, we'd like to stress that the biogeochemical module has been extensively calibrated within BALTSEM model, plausibly reproducing ecosystem dynamics in the entire Baltic Sea (e.g. Meier et al., 2018), from the cold, annually ice-covered, almost fresh, and severely P-limited Bothnian Bay (i.e. very much Onego-like), to the warmer, mesotrophic Gulf of Finland and the Kattegat with a single set of parameterizations and constants in both basin-wise horizontally averaged and true 3D versions (Gustafson et al., 2017; Ryabchenko et al., 2016, Isaev et al., 2020). Besides, similar formulations have already been favorably tested at Lake Ladoga (Isaev and Savchuk, 2020). Such simultaneous coverage of a wide range of ecological conditions makes us somewhat confident in application of largely the same set of formulations to Lake Onego.

Unfortunately, the presented manuscript creates a wrong impression that we fully avoided the calibration. As can be seen from SPBEM formulation (Table A1 in Isaev et al., 2020), the major difference between "cyanobacteria" and "summer species", given in parameterization, is the capability to fix molecular nitrogen under appropriate conditions. Without such conditions both variables behave almost identical as was known from the Bothnian Bay and Lake Ladoga simulations and was demonstrated by the initial runs for Lake Onego. Therefore, as indicated in the text (lines 87-93), we excluded the "diazotrophic cyanobacteria" group as a separate variable, thus actually merging such other ecosystem functions and biogeochemical fluxes as nutrient uptake, mortality, sinking, etc., into "non-diatoms" group. Such adaptation requested recalibration

necessary also to better separate dynamics of cold-water diatoms from summer "non-diatoms". As was also shown by the initial test runs, all the other temperature- and concentration dependent processes, being already calibrated for similar conditions in the Baltic Sea and Lake Ladoga, have not requested urgent re-calibration, unsupported by sufficient amount of contradicting reliable observations, and were left as they were. Based on these considerations we will revise the text situated at lines 90-93, as follows (and add a Table A1 in Appendix, if requested by the Editor):

**"Thus, autotrophs were presented by only two variables, diatoms and non-diatoms, that comprised all the other (summer) phytoplankton species, for example, chlorophytes, chrysophytes, and cyanobacteria. Such reformulation requested recalibration of several phytoplankton parameters, necessary also to better separate dynamics of summer "non-diatoms" from cold-water diatoms (Table A1). As was also shown by several test runs, all the other formulations, being extensively calibrated and tested in similar temperature and trophic conditions (e.g. Gustafsson et al., 2014; Isaev et al., 2020 and references therein) have not requested further fine-tuning in the absence of abundant and reliable contradicting observations."**

We also suggest to add Table A1 (see below), in order to update a set of constants from Isaev et al., 2020 (with actual values used for Lake Onego)

2. validation: there is really too little data for the model to be properly validated. Comparing a few simulated values on Lake Onego with those measured on other "similar" lakes is not sufficient for this. Yet, the authors could have considered some remote sensing measurements issued from satellite images that would have help them a lot for this validation process.

Appropriate model validation has always been our concern for decades (e.g. Savchuk and Wulff, 1996; Savchuk, 2002; Savchuk et al., 2012, Gustafsson et al., 2017; Isaev et al., 2020), especially with a wealth of data available for the Baltic Sea. Unfortunately, the situation is drastically different with Lake Onego

Here, we respond to the comment on field data, while possibilities of remote sensing are considered further down below.

Our mechanistic model is based on a mass balance approach, describes internal biogeochemical cycles and accounts for external sources and sinks (imports and exports), either prescribed as forcing functions or computed according to formulations. Consequently, both all the simulated fluxes and concentrations resulting from their interplay are strongly deterministically coupled and thus, confined. Therefore, their reliability should be judged by a *simultaneous* fitting of many fluxes and concentrations in the known ranges reported for both Onego and similar boreal oligotrophic lakes. For example (our Table 3), the nutrient sedimentation of OM cannot be very much higher than nutrient uptake during primary production of OM simulated with the plausibly given phytoplankton specific growth rates (note good PP validation, see suggested Table 1and Fig. 7 below). Similarly, the sediment release (and denitrification) of nutrients cannot be order of magnitude, or even several-fold higher or lower, thus causing (unreported? unobserved?) fast accumulation or depletion of sediment nutrients. The plausibility of simulated rates is estimated by a comparison to sediment rates from similar environments (our lines 443-465). This can be said about all the other processes in Fig. 1 and Table 3. That's why we consider the quantitative information from other lakes combined with Onego data as quite relevant and justifying.

Initially, the main approach was to compare simulation to information already published as tables, graphs, and maps in books, atlases, and papers (appropriately referred to in the manuscript), thus leaving responsibility of interpretation of scarce data to corresponding authors-experts. However, your request made us to additionally dig up some information, just to indicate that there are only a few dozen of measurements irregularly scattered over two decades, mostly only in summer (see Tables A2 and Fig. 6 and 10 below).

Therefore, we prefer to talk about typical ranges rather than calculate some highly uncertain (or even mathematically incorrect) statistics, as we already explicitly admitted at L157-158.

A recent paper by Galakhina et al. "Current chemistry of Lake Onego and its spatial and temporal changes for the last three decades with special reference to nutrient concentrations" is just pre-printed in "Environmental Nanotechnology, Monitoring and Management" (https://doi.org/10.1016/j.enmm.2021.100619) and presents results of 3 (three) surveys in September 2019, June, and August 2020 (that is beyond our simulation interval) at 35 stations in different regions of Lake Onego as well as some older scattered data from 1992-2018. The samples were taken only from surface and bottom layers. Note, that we intend to use this paper and information from it in a possible revision of our manuscript at L 230:

**"Such simulated quasi-stability of TP concentrations and clearly decreasing DIN concentrations (Fig. 8 c, e)) is validated by the recent field surveys. Galakhina et al. (2021) found at the surface of pelagic part of Lake Onego a statistically significant decreasing trend of the DIN:TP weight ratio from 33.7 in 1992-1995 to 23.7 estimated from the field surveys made in September 2019, June and August 2020, which is well comparable to a simulated decrease of DIN:TP ratio from 36.5 ± 1.9 in 1992-1995 to 21.2 ± 1.1 in 2011-2015; these mean ± S.D. values are computed for the surface layer in I-VI limnic areas (gf. Fig. 2) over biological summer (cf. Figs. 7 and 11 a)."**

and 250-255, most likely enriching the text with several numbers from this paper

3. simulation: the authors made only one simulation instead of performing a model exploration that could have provided some estimation of the uncertainties on the model outputs. Indeed, the author says that the simulation results are plausible but nowhere they give an estimation of this "plausibility" (and so the uncertainties) of the results.

As we indicated above, we have made several test and calibration runs, which are not suitable for the uncertainty estimates in the sense indicated by you. Also, we are pretty familiar with- and confident in the biogeochemical module behavior from over two decades of its exploitation (referred to in the text), as hopefully, are many our readers. Therefore, the formal sensitivity analysis has never been considered as a goal of this study and we have not tried to artificially and unnecessarily alter constants. Also, we intentionally used here the word "plausibility" (rather in a sense of G. Polya "Mathematics and plausible reasoning") instead of stronger and more certain "reliable", "realistic", "accurate" and such (again L157-158).

3. conclusions: the authors made a lot of comments and conclusions, as if the simulations they performed were reliable. Moreover, they argue that the simulated results can be used as a form of re-analysis when there is almost no data available.

As was already indicated above and explicitly said in Conclusion 2 (L473-476), our study "…led us to *believe in plausibility* of simulation." For the lack of anything based on observations, we attempt to provide the diverse scientific community (from hydro- and geochemists to hydro-biologists and environmentalists) with plausible 30-year hindcast of the Lake Onego biogeochemistry that they can never obtain otherwise but can now analyze and discuss the simulated fluxes and concentrations comparing it to their limited data and perceptions. Interesting, that the spring bloom case (Conclusion 3) has already made local biologists insistently look for the possibility to make measurements in the smelting ice phase (see our Figs. 4, 7-9), unrealized so far, unfortunately.

Finally, a lot of comments and information are given but sometimes the most essential ones are missing. In particular, with regard to the available data, which is of importance here, details are not always given. The

comparisons are not well explained and the value of the errors between simulated and observed data are not given.

Unfortunately, although we do not quite understand such general comments, we'll try to answer here also generally, with more details in the Specific comments Section.

We are neither in a position nor have any intention compiling such original data (with all its proprietary and copy rights) in our manuscript instead of its generalization with appropriate references. Neither it is our responsibility and capability to perform an enormous task of creating such an ecosystem database from over decades of research.

Following the usual practice of substantiating our own judgments and statements, including numbers, we tried to give appropriate references to all the quantitative and qualitative information, which we used for comparison, trusting the sources and expecting similar trust from the readers to our compilations and references. The only further details, which we could provide are specific indication of graphs, maps and tables (but certainly not copying them in our manuscript) in the referenced sources. Just for example, the references at L285 "…sediment layer of 5 cm thickness (Filatov, 2010; Kalinkina and Belkina, 2018) …" could be edited to "…Filatov, 2010, p 101-102 (there are no Figs numbers with the legend), Kalinkina and Belkina, 2018, Table 1). This original information is given as CC 2-5 below, while our generalization of these data is contained in Table 1 of the manuscript. We also do hope, that with such compilations we are making some additional information available for non-Russian reading audience.

Despite common terminology, we also prefer not to consider the differences between occasional measurements made here and now, and numbers simulated within certain time-space window as errors (=mistakes) of either measurements or modeling, just discrepancies, whose nature deserves and sometimes gets further analyses (e.g. L188-191). In case of revision we'll carefully check and use such possibility further.

I understand that this case study is complicated, because of the lack of observation data. Models are obviously interesting tool that we must use, but in combination with observation data. Without them, it is impossible to validate the model outputs and to make conclusions.

We still believe, that within: a (A) system and (B) mass balance approach with (C) all the fluxes and concentrations being deterministically coupled and, thus, pretty much confined, the simultaneous fitting of them in the know observed ranges is an important indication that the model thirty-year hindcast of seasonal dynamics is non-contradictive and plausible enough to serve as a valuable addition to the sparse data and be useful for the diverse scientific community in studying the Lake Onego biogeochemistry, comparing this reanalysis with their knowledge and perceptions.

According to me, satellite images should be the first thing to work with when direct measurement data are not available. In the case of lake Onego, which is moreover large, this will be all the easier. The second thing is to do model exploration to draw conclusions from a set of simulations rather than one.

We consider "remote sensing" below (com. L170) after answering to comments about hydrophysics (L164-165)

Finally, if so little data are available, considering a 1d vertical model could be a first interesting step.

In terms of hydrophysics, seasonally stratified Lake Onego, with its baroclinic Rossby radius of deformation RR smaller by several orders than the lakes' horizontal dimensions (Rukhovets and Filatov, 2010), could hardly be treated as horizontally homogeneous (e.g. our Fig. 8). Besides, *the 1D vertical model*, similar, for instance, to Flake and FLake-Eco models (http://www.flake.igb-berlin.de;, indicated as not fitting for the

purpose at Lake Ladoga by the authors: S.A. Kondratyev, M.V. Shmakova, S.D. Golosov, I.S. Zverev, K.D. Korobchenkova. Modeling in Limnology. Experience of IL RAS. Gidrometeorologiya i Ekologiya. Journal of Hydrometeorology and Ecology. 2021, 65: 607—647. [In Russian]. doi: 10.33933/2713-3001-2021-65-607-647), *would not make up by itself for the lacking data anyway* and could not be used for the exploration of not only the lake-scale long-term ecosystem changes but also such applied spatial problems as localization of the fish farms, water consumption intakes or wastewater outputs in the conditions of socio-economic and climate changes, which are the main ambitions for this model. Note that we intend to express such ambitions in the new ending of Introduction by replacing L73-75 with the following:

**"The avoidance of such biases is especially important for our intention of producing prognostic estimates.**

**Thus, the main purpose of this paper is a presentation of the 3D ecosystem model capable to a certain extent fill the historical deficit in observations of nutrient variables and, especially estimates of the biogeochemical fluxes. According to one of the major functions of simulation modelling, we intend implementing this model as a complimentary form of studies of Lake Onego ecosystem, providing a unifying formal platform for testing and discussing consistency of both model parameterizations and results of hydrological, hydrochemical, hydrobiological, and geochemical research. Furthermore, the model will be implemented as a major tool for a wide range of projections, from applied tasks of localization of fish farms, water intakes, and wastewater outlets to long-term large-scale ecosystem evolution under different scenarios of climate change and socio-economic development."**

**Specific comments**

section 2.1: in the section "model presentation", the author said that the model they consider is the SPBEM. They give a reference (Isaev et al. (2020)) to the reader in which, according to them, all the equations, parameters, constants, etc are given. And they explain what adaptation they made to apply the model on the case of Lake Onego. If I well understand, they change a little bit the structure of the biogeochemical model, but they keep the same parameters values than those used (and calibrated) to simulate the biogeochemical functioning of Baltic Sea. How can the authors justify that? We know that the models can be very sensitive to parameter values, that the parameters values can be different from one ecosystem to the other, which is the reason why the calibration step is important. I understand that the author do not have a lot of available measurements, but this is not a sufficient reason not to pay attention to the parameter set used in the model.

See our answers and considerations above, under 2. Calibration and further

l 90-91: in the modified version of the model, only two variables were considered for the autorotrophs: diatoms and non-diatoms. How the authors have chosen the values of the parameters corresponding to the non-diatoms group that gather several variables of the original SPBEM?

By the past experience. Generally, we have used such variable as a thermophilic phytoplankton functional group (PFG) under different names as "summer species", "small summer species", "flagellates", etc. for decades (e.g. Isaev et al., 2020, TableA1), and somewhat tuned constants in its parameterization (suggested Table A1). See also our answer under 2. Calibration.

l 113-115: for the 40-year spin-up simulation, did the authors consider some nutrients inputs from the river? If so, it could have led to some accumulation of the nutrients in the sediments that is really slow, no?

Exactly! 40-year spin up full-scale simulation for the development of initial conditions has been made with the repeating (cycling) full set of the boundary conditions, including external nutrient inputs (river, atmosphere, coastal sources). Indeed, it has led to some accumulation of nutrients, which is why it took

about 40 years to achieve a kind of steady state of the sediment nutrients, corresponding to conditions of 1984. If requested, we could slightly expand the main text as follows:

**The initial conditions for the hindcast simulation of the biogeochemical dynamics of Lake Onego ecosystem between 1985 and 2015 were generated during a spin-up simulation with the boundary conditions, including external nutrient inputs, repeated for 1984. With such repeating forcing, the quasi-steady state of seasonal dynamics was reached in 40 years, mainly because of a slow evolution of sediment nutrients.**

l 150, section 3: which observation data are available exactly? A table that summarizes all the available data that have been used for this study would be helpful.

We have used both qualitative and quantitative information from a couple of dozens referenced books, papers and atlases, including graphs, maps, and numbers given in Tables and, as already explained above, never had intended to make any inventory of the raw original data. However, in response to your request, we were able to obtain some additional data on temperature and PP, which could now be presented in new Table 1 and Figs 6 and 10 (see below), together with the indication of (sparse) data availability.

l 164: what do the authors mean by "we omitted the analysis"? Have the authors access to some other measurement data that they did not consider? Or did the authors only show the ice cover and the water temperature because it is the only measurement data they have?

Perhaps, it is just a confusing imprecise statement, by which we meant that we had no intention to validate the simulated hydrophysics, for which there are much more publications but which is quite a different separate task (see below), and focused instead on those features that are most important for the seasonal dynamics of biogeochemical cycling and could be compared to some available estimates based on observations.

Also, we bear in mind that 30-year atmospheric forcing based on the ERA-Interim reanalysis fields (https://www.ecmwf.int) with a basic spatial resolution of 80 km (based on a comparatively sparser network of meteorological stations in this region) is hardly suitable for reliable sequential reproduction of transient synoptic situations of 5-7 days duration. Therefore, we did not attempt the detailed pair-wise comparison of 532 measurements made in June–August over 15 years of 1992–2007 (L180), *a priory* considering it confusing and misleading rather than validating. Instead, we estimated average summer temperatures, both observed and simulated, as indication of plausibility of the simulated summer thermic situation.

l 165: the authors says that "surface water temperature" is an "important integral indicator of the hydro-thermodynamics" which I do not agree with. Surface water temperature is highly influenced by external meteorological inputs and does not reflect the complex thermal structure of lakes, in particular the stratification periods that play a key role in ecosystem dynamics.

We both agree and disagree. Of course, the surface water temperature, occasionally measured here and now, is highly influenced by the synoptic atmospheric variations and internal small-scale water movements (cf. L187-191). On the other hand, the seasonal dynamics of the surface water temperature reflects a complex development of the convective spring and autumn turnover with a formation of summer stratification in between (see Fig. 9). That's why we consider a steady reproduction of such dynamics in a long-term simulation at both the entire lake scale (L179-185) and for the three offshore stations (Fig. 5) as indicator of its plausibility. To further validate it we suggest an addition of Fig. 6 (see below) with vertical temperature profiles with appropriate short explanation somewhere at L180-190 and L361-375. Similar considerations concern also the integral seasonal dynamics of the ice cover percentage. Note, that because of both

interannual and vertical irregularity of sparse measurements we are satisfied with just an average vertical distribution, considering any dispersion estimates futile.

l 170: why the authors did not used the entire satellite images for comparison with the simulated field? This is one interesting advantage of using a 3D model? Moreover it gives access to additional data that are of particular interest in a case such as this one, where only a few measurements data are available.

The remote sensing could in general be used for validation of hydrophysics (ice, surface temperature, water level, etc.) or some environmental characteristics (water color, suspended matter, chlorophyll, etc.). As we had never intended in this study an extensive validation of hydrophysics, we have not attempted such a huge job, extremely complicated also by the inevitable discrepancies between long-term simulations forced by atmospheric reanalysis based on an 80-km grid and comparatively poor and uncertain satellite coverage due to abundant cloudiness and ice (Filatov et al., 2019). However, since the simulated spring bloom in the model starts during melting ice season, we find it is important to validate our reasonable simulation of the seasonal dynamics of the average ice coverage (Fig. 4).

As for the environmental characteristics (simulated by the model), we could have tried some satellite chlorophyll data (e.g. Kahru et al., 2016). But we would not do it, at least, for two reasons. Fundamentally, however important are synchronous chlorophyll fields obtained by remote sensing as a *relative measure* of phytoplankton biomass and its dynamics, they cannot be reliably used for estimation of in situ biomass without special empirical conversions, localized both in space and time, accounting for an order of magnitude seasonal variation of C:Chl ratio, from about 20-30 in spring to 200-300 g C : g Chl-a in summer (cf. Meier et al., 2018 and references therein). Furthermore, in contrast to Lake Ladoga (Pozdnyakov and Filatov. Interannual water quality variations in Lake Ladoga in spring during 2016 and 2017: satellite observations. Fundamentalnaya i Prikladnaya Gidrofizika. 2021, 14, 1, 79–85. doi: 10.7868/S2073667321010081 (in English); Morozov et al., Long-term phenology of water quality parameters in Lake Ladoga: A satellite-based study, submitted to JGLR), the specific retrieval algorithm for Lake Onego Chlorophyll has not been developed yet. Some pilot attempts (CC6) could hardly be used for any reliable analysis, the least, validation.

l 173, figure 4. Were the data of this figure used for a calibration procedure or not?

No, they were not. As correctly noted by Reviewer 2, both Fig. 4 and 5, and now also new Fig.6 have been produced long after the simulation. So, this is a real post-simulation validation.

l 195: the authors should explain briefly somewhere in the manuscript what is the concept of "biological summer" because they refer to it several times.

The concept does not belong to us and was borrowed from the referenced sources (L197-198). For us, it seemed self-evident that BSD has everything to do with the summer stratification division of the water column on epi- and hypolimnion. We do not want complicate the text with more detailed retelling of the conception and we used BSD here simply because there are some published numbers that we used for comparison. What we could do in the possible revision is to add a reference to our Fig. 9a and Fig. A2.

l 204-206: The authors mention a difference (based on field data) in the BSD in the shallow areas and in open deep waters. Did they retrieve this difference in the simulations? Can they give some estimations based on the simulations that can be compared with the observed BSD?

We will add a sentence after "Based on sparse field data, BSD in the shallow areas (100–110 days) was markedly longer than that in open deep water (85–90 days) (Efremova et al., 2016)." (L204-206): **"In**

**simulation, BSD in the area shallower 30 m was 99 +/-10 days comparing to BSD in deeper areas with 93+/-10 days"**

paragraph 3.1.2 The authors show some simulation results but there is no data available for comparison. How can we trust such results? The only comparison that are made is on the annual integrals of the simulated phytoplankton primary production that is compared with values of other lakes...

We will add a paragraph in 3.1.3 with comparison of summer PP measurements with simulation, that show pretty good comparability, referring to new Fig. 10 and Table 1.

l 223-224: the simulations show, for each year, a strong phytoplankton spring bloom and a minor autumn bloom. Was that also observed in reality? Here again satellite data could be useful.

The entire point is that the strong spring bloom has not been considered by biologists of NWPI  KRC RAS in their phenological descriptions due to the lack of observations for the corresponding period. Perhaps, satellite data could show the fact of the bloom, at least, qualitatively, but are highly uncertain even for Lake Ladoga (see above Pozdnjakov and Filatov, 2021; Morozov et al., submitted,), while for Lake Onego such data do not exist.

l 252-254: do the authors speak about simulated results or observations? it is not clear.

We will add word **"… *simulated* annual primary production**… at L252

l 256: can we get more information about these direct measurements?

l 257-258: why the author compares a range of observed values ("from 413 mg C m-2 day-1 at the top of Petrozavodsk Bay to 122 mg C m-2 day-1 in the open areas of the Bay") with an average simulated value on the entire limnic area?

We'll answer for both these questions by new Table 1 and Fig. 10 with the appropriate explanations.

l 276: what does "in good agreement with the phenology of the Lake Onego ecosystem deduced from field observations" mean? the author should explain which field observation they are talking about and make a detailed comparison.

These phenological features, presented at L271-275, are described in publications based on generalizations of field observation and referenced at L277, including those written in English. As an example, see CC7-8, a detailed description of which, that would just repeat the published source, could hardly be appropriate in our manuscript. Besides, the phenology is also described, discussed, and as much validated as we could in Section 3.2 Seasonal dynamics.

l 284: here again give details about the available measurements? is some measurement at one day? several times? what was measured, etc...

More precisely, we will replace "measurements" with "**estimates**", while these estimates are based on referenced publications and already exemplary explained above and in CC2-5

Table 1: why did the authors give a range of values for the measurements and a mean and standard deviation for the simulated results? what does this range of values represent for the measurements?

As was explained above and in CC2-5, the ranges were eye-balled from maps in Filatov (2010) and estimated from irregular measurements scattered across years in Table 1 in (Kalinkina and Belkina, 2018), while from the simulated regular time-series we were able to calculate SD.

l 341: which reported maximum? is it a field observation?

"…the *field observations* (Tekanova and Syarki, 2015). The concurrently developing *simulated* non-diatom complex reached its maximum biomass of 0.05 g ww m-3 in July–August, which is close to the *reported* maximum of 0.034 g ww m-3." For clarity, we could add "… reported **observed**…"

paragraph 3.2: here again, the authors make a lot of comments based on the simulations whereas there is no data for comparison. Therefore, the conclusions they made (l 355-360) are unreliable.

"Unreliable" - not able to be trusted or believed (Cambridge Dictionary); unable to be trusted to do or provide what is needed (Merriam-Webster).

We strongly disagree about reliability, for the reasons already explained above. The entire point is that there are no observations at all that would cover necessary seasonal time window. On the other hand, there is absolutely no reason for the absence of the spring phytoplankton bloom in Lake Onego developing according to mechanisms described in the text (Section 3.2) and bearing the characteristics like in other lakes referred to in the text. That is why it is important to quantitatively present such phenomenon for further discussions and, hope for future observations that would somehow be made in the melting ice on the vast open areas.

As stressed at L361-362 "The mechanisms of seasonal dynamics in boreal lake ecosystems are well known, although the under-ice and melting-ice phases have often been overlooked (Hampton et al., 2015)." Here we just illustrate such mechanisms in detail with an aid of Fig. 9, thus also verifying the plausibility of simulation. In the possible revision of the manuscript we'll add here reference to vertical temperature profile (new Fig 6), as well as some numbers on phosphorus concentrations taken from Galakhina et al. mentioned above.

paragraph 3.3: same remark than for paragraph 3.2.

The same answer as above, with the simulated integrals and rates justified ("validated") by the *simultaneous* plausible fitting of all the concentrations and fluxes in the ranges reported for Lake Onego and relevant locations elsewhere.

technical correction

line 61: reproduces instead of reproduced

Done, thank you

line 66-68: there is something missing in this sentence. "can be used" but for what?

We could expand it as "…**can be used for further analysis together with existing knowledge on the relationships between other, not simulated ecosystem components…**"

lin 85: there are two references Isaev et al (2020). Add a a and b to distinguish the two papers.

The references are Isaev and Savchuk (2020) and Isaev et al., (2020), i.e. different

line 86: "those our formulations"? put "these formulations" instead

We'll correct to **"…these our formulations…"** to extra stress the authorship

line 89: lakes instead of lake

We mean one lake, namely Lake Onego

line 94: "and thus never became limiting" instead of "that is never became limiting"

Done, thank you

line 164: "we omitted the analysis" instead of "we omitted analysis" Done, thank you
* * *
New table and figures

[Figure]

New Figure 6. Observed (red) and simulated (blue) average vertical distribution of the water temperature in three limnic areas of Lake Onego (cf. Fig. 2 and (old) 9a), showing also time coverage and amount of observations. Note differences in scales.

New Table 1 –Phytoplankton primary production in Lake Onego according to observations (Rukhovetz and Filatov, 2010, Table 1.4) and simulation for the period from 1989 to 2006, mg C m$^{-2}$ day$^{-1}$.

|  | Data | Model | | |
|---|---|---|---|---|
|  | summer | May – Oct | Jun - Sep | Jul – Sep |
| Southern Onego | 88.3 ± 15.5 | 82.4 ± 60.7 | 72.0 ± 48.8 | 53.6 ± 4.7 |
| Central Onego | 96.3 ± 10.5 | 121.5 ± 109.7 | 111.2 ± 105.2 | 69.0 ± 15.4 |
| Petrozavodsk Bay |  |  |  |  |
| top part | 412.9±62.7 | 350±121 | 491±131 | 494±86 |
| central part | 199.8±38.3 | 254±56 | 297±61 | 284±44 |
| outer part | 122.3±21.7 | 180±57 | 166±37 | 151±25 |
| Kondopoga Bay |  |  |  |  |
| central part | 286.7±24.2 | 221±90 | 324±93 | 288±76 |
| outer part | 217.4±23.3 | 153±71 | 185±81 | 134±26 |

[Figure]

New Fig. 10. Model-data box-and-whisker plots of the phytoplankton primary production (mg C m$^{-2}$ day$^{-1}$) in different limnic areas of Lake Onego (cf. Fig. 2). Note paucity and irregularity of observations.

Suggested Appendix

**Table A1.** Parameters for the autotroph groups.

| Symbol | Parameters | Units | Diatoms | NonDiatoms |
|---|---|---|---|---|
| $\lambda_{AN}$ | N/P ratio | mgN/mgP | 7 | 7 |
| $a_{gi}$ | Maximum growth rate at 0 °C | day$^{-1}$ | 1.25 | 0.75 |
| $b_{gi}$ | Temperature constant for growth and mortality | °C$^{-1}$ | 0.078 | 0.12 |
| $I_{0i}$ | Optimal photosynthetically active radiation | W m$^{-2}$ | 25 | 50 |
| $h_{Ni}$ | Half-saturation constant for inorganic nitrogen | mg N m$^{-3}$ | 7.0 | 3.5 |
| $h_{Pi}$ | Half-saturation constant for phosphate | mg P m$^{-3}$ | 1.5 | 1.5 |
| $c_{rN}$ | Threshold ammonium concentration | mg N m$^{-3}$ | 21.0 | 21.0 |
| $a_{mi}$ | Mortality rate at 0 °C | day$^{-1}$ | 0.4 | 0.15 |
| $b_{mi}$ | Temperature constant for mortality | °C$^{-1}$ | 0.063 | -0.2 |
| $a_{si}$ | Sinking velocity at 0 °C | m day$^{-1}$ | 0.5 | 0.1 |
| $\gamma_{mi}$ | Mortality rate adjustment for "fit" conditions | | 4 | 4 |
| $\gamma_{si}$ | Sinking velocity adjustment for "fit" conditions | | 4 | 4 |
| $\alpha_i$ | Availability as food source | | 1.0 | 1.0 |

Clarifying considerations (CC)

CC1. Availability of the phytoplankton primary production measurements in the different limnic areas of Lake Onego. We show generalization of this information in new Fig. 10

| | | Number of measurements | | | | | |
|---|---|---|---|---|---|---|---|
| | | Month | | | | | |
| Region | Years | Jun | Jul | Aug | Sep | Oct | All |
| BigOnego | 1989 - 2010 | 8 | 5 | 9 | 3 | 3 | 28 |
| CentralOnego | 1994 - 2010 | 17 | 6 | 13 | – | – | 36 |
| SouthOnego | 1994 - 2005 | – | 3 | 5 | – | – | 13 |
| PetrozavodskBay | 1989 - 2010 | 27 | 11 | 13 | 6 | 3 | 60 |
| KondopogaBay | 1989 - 2010 | 22 | 12 | 34 | 14 | 11 | 93 |

CC2. Distribution of the sediment types (granulometric composition at p. 100 in Filatov (2010). (left)

[Figure]

[Figure]

CC3. Distribution of the C-org (left), TP (middle) and N-org (right) in the Onego sediments at p. 101 (Filatov, 2010). (right)

Калинкина Н. М., Белкина Н. А. Динамика состояния бентосных сообществ и химического состава донных отложений Онежского озера в условиях действия антропогенных и природных факторов // Принципы экологии. 2018. № 2. С. 56–74. DOI: 10.15393/j1.art.2018.7643

[Figure]

Рис. 1. Расположение станций отбора проб ДО и бентоса. 1 – зона интенсивного антропогенного воздействия; 2 – буферные зоны заливов; 3 – глубоководные участки

Fig. 1. Location of sampling stations of bottom sediments and benthos. 1 – zone of intensive anthropogenic impact; 2 – buffer zones of the bays; 3 – deep water areas

CC4. Sampling sites in (Kalinkina and Belkina, 2018)

Калинкина Н. М., Белкина Н. А. Динамика состояния бентосных сообществ и химического состава донных отложений Онежского озера в условиях действия антропогенных и природных факторов // Принципы экологии. 2018. № 2. С. 56–74. DOI: 10.15393/j1.art.2018.7643

Таблица 1. Химический состав донных отложений Онежского озера (слой 0-5 см)

| Период | № ст. | Eh, мВ | pH | C | ППП | N-NH₄ | N | Fe | Mn | | |
|---|---|---|---|---|---|---|---|---|---|---|---|
| | | | | | | % от сухой навески | | | | | |
| Кондопожская губа | | | | | | | | | | | |
| 1991-1999 | К_3 | 89 | 6,53 | 19,8 | 45 | 0,228 | 0,70 | 1,6 | 0,34 | 0,13 | 0,21 |
| 2000-2005 | | 11 | 6,23 | 16,8 | 42 | 0,010 | 0,63 | 3,1 | 0,19 | 0,14 | 0,19 |
| 2005-2010 | | 282 | 6,19 | 26,8 | 56 | 0,048 | 1,01 | 3,1 | 0,28 | 0,18 | 0,27 |
| 1991-2000 | К_4 | 60 | 6,80 | 5,1 | 9 | 0,007 | 0,51 | 0,8 | 0,09 | 0,05 | 0,10 |
| 2000-2005 | | 68 | 6,88 | 13,1 | 25 | 0,057 | 0,47 | 3,1 | 0,27 | 0,11 | 0,16 |
| 2005-2010 | | 77 | 6,57 | 12,4 | 33 | 0,073 | 0,60 | 3,6 | 0,14 | 0,15 | 0,21 |
| 2010-2015 | | 138 | 6,24 | 13,7 | 29 | 0,008 | 0,61 | 3,0 | 0,17 | 0,15 | 0,20 |
| 1991-1995 | К_6 | 151 | 6,77 | 7,2 | 27 | 0,046 | 0,72 | 3,2 | 1,08 | 0,18 | 0,27 |
| 2000-2005 | | 247 | 6,67 | 7,1 | 21 | 0,063 | 0,48 | 7,0 | 1,07 | 0,25 | 0,31 |
| 2005-2010 | | 73 | 6,73 | 7,2 | 18 | 0,004 | 0,20 | 6,6 | 0,47 | 0,24 | 0,28 |
| 1990-1995 | К_7 | 301 | 6,57 | 2,6 | 14 | 0,022 | 0,62 | 0,6 | 0,43 | 0,07 | 0,13 |
| 2001-2005 | | 366 | 6,44 | 4,4 | 17 | 0,079 | 0,30 | 5,6 | 0,99 | 0,15 | 0,22 |
| 2005-2010 | | 374 | 6,32 | 4,4 | 10 | 0,003 | 0,15 | 5,1 | 1,00 | 0,13 | 0,21 |
| Петрозаводская губа | | | | | | | | | | | |
| 1990-1995 | Р_5 | 112 | 6,59 | 1,7 | 4 | 0,020 | 0,20 | 0,9 | 0,06 | 0,06 | 0,11 |
| 2000-2005 | | 290 | 6,99 | 3,5 | 9 | 0,018 | 0,19 | 3,5 | 0,36 | 0,13 | 0,15 |
| 2005-2010 | | 335 | 5,63 | 3,8 | 10 | 0,002 | 0,11 | 4,2 | 0,23 | 0,15 | 0,16 |
| 1990-1995 | Р_2 | 179 | 6,59 | 2,8 | 13 | 0,021 | 0,33 | 1,2 | 0,93 | 0,12 | 0,16 |
| 2000-2005 | | 419 | 6,70 | 3,9 | 9 | 0,019 | 0,31 | 5,6 | 1,19 | 0,15 | 0,20 |
| 2005-2010 | | 582 | 6,54 | 3,8 | 13 | 0,003 | 0,29 | 5,4 | 1,43 | 0,18 | 0,23 |
| 2010-2015 | | 575 | 6,02 | 4,4 | 16 | 0,004 | 0,39 | 5,0 | 1,72 | 0,09 | 0,22 |
| 1990-1995 | Р_3 | 347 | 6,59 | 2,9 | 12 | 0,021 | 0,66 | 1,3 | 1,37 | 0,12 | 0,16 |
| 2000-2005 | | 332 | 6,78 | 4,1 | 12 | 0,020 | 0,26 | 4,7 | 0,48 | 0,16 | 0,21 |
| 2005-2010 | | 353 | 6,08 | 2,6 | 10 | 0,002 | 0,13 | 4,9 | 0,66 | 0,12 | 0,17 |
| 2016* | | 472 | 5,86 | 4,4 | 15 | 0,003 | | | | | |
| Большое Онего | | | | | | | | | | | |
| до 2000 | В_1 | 357 | 6,55 | 3,6 | 15 | 0,041 | 0,67 | 1,9 | 1,25 | 0,11 | 0,15 |
| 2000-2005 | | 408 | 6,23 | 4,6 | 16 | 0,023 | 0,28 | 7,3 | 1,48 | 0,15 | 0,22 |
| 2005-2010 | | 429 | 6,46 | 4,2 | 18 | 0,013 | 0,34 | 7,3 | 1,48 | 0,15 | 0,27 |
| 2016 год | | 569 | 6,70 | 4,3 | 17 | 0,008 | 0,57 | | | 0,10 | 0,21 |
| 2000-2005 | В_2 | 341 | 6,14 | 4,9 | 13 | 0,060 | 0,30 | 4,7 | 1,01 | 0,13 | 0,21 |
| 2005-2010 | | 508 | 6,41 | 2,3 | 14 | 0,003 | 0,39 | 5,2 | 0,92 | 0,12 | 0,18 |
| 2016* | | 656 | 6,54 | 4,7 | 18 | 0,003 | 0,50 | | | 0,11 | 0,19 |
| Центральное Онего | | | | | | | | | | | |
| 1990-1995 | С_1 | 316 | 6,69 | 3,2 | 17 | 0,039 | 0,55 | 2,5 | 1,81 | 0,15 | 0,31 |
| 2000-2005 | | 515 | 6,41 | 3,5 | 17 | 0,015 | 0,32 | 7,2 | 1,3 | 0,22 | 0,29 |
| 2005-2010 | | 344 | 6,40 | 2,9 | 14 | 0,002 | 0,20 | 7,6 | 0,82 | 0,27 | 0,34 |
| 2010-2017 | | 459 | 6,61 | 2,9 | 13 | 0,003 | 0,30 | 3,4 | 0,98 | 0,08 | 0,28 |

Примечание: * – единичные пробы.

CC5. Table 1. Chemical composition of the Lake Onego bottom sediments (layer 0-5 cm) (Kalinkina and Belkina, 2018)

[Figure]

Май–июнь Июль

Август Сентябрь

algal_2_median,[mg/m^3]

0.0 1.0 2.0 3.0 4.0 5.0

**CC6.** Test interpretations of remotely sensed chlorophyll-a in Lake Onego in 2011.

[Figure]

Fig. 2. Multiyear seasonal dynamics of the primary production process and abiotic parameters. (*I*) PP, (*2*) surface water temperature, (*3*) PAR, (*4*) N-NO$_3$, (*5*) P$_{total}$.

[Figure]

Fig. 3. Multiyear seasonal dynamics of the primary production process and biotic parameters. (*I*) PP, (*2*) diatom phytoplankton biomass, (*3*) biomass of "nondiatom" phytopankton, (*4*) chlorophyll *a*.

CC7. Note that all curves are normalized by a maximum value found on the presented time span and start only in the end of May, when normalized diatom biomass inexplicably goes steeply down from apparently much higher values (from the spring bloom revealed in our simulation, as we interpret the dynamics). Fig.3 also gives a false impression of summer dominance of (3) due to normalization with maximum value, whereas in reality diatoms dominate by biomass over the entire summer

[Figure]

Распределение общего фосфора
в поверхностном слое воды летом

Распределение общего фосфора по разрезу А–А,
мкг/л

CC8. Summer distribution of TP in the surface layer and cross-section AA in *spring* (*5-11 of June* (sic!)) 1995, summer 2001, and autumn 1992. Note paucity and irregularity of surveys.

---

## Author Comment (AC2)

**REVIEWER 2**

Review of: Modeling of the large-scale nutrient biogeochemical cycles in Lake Onego.

I previously reviewed this paper when it was submitted to Limnology & Oceanography. Of interest to me was to understand if the authors had responded to my earlier comments, the comments of another reviewer and the handling editor. These comments should have translated to changes throughout the paper. In the first instance, I noted the Abstracts of the two papers (L&O and Biogeosciences) were identical. I have gone through the reviewers' comments from the earlier review (in blue) and added new text (in black) that reflects whether I consider the earlier comments are adequately dealt with in the current submission to Biogeosciences.

Thank you for agreeing to review the revised manuscript again, which, in contrast to L&O case, gives us a chance for explanations, discussion, and revision. Excuse us, please, for sometimes kind of didactic tone in reminding certain basics of mechanistic modeling, which we had to keep reminding time and again in discussions even with some of our fellow modelers. To avoid incontinences with multiple attachments, we structure this single PDF in the following succession: 1) Full replays and explanations to your Review; 2) suggested new comparisons with measurements of vertical temperature distribution (new Fig.6) and primary production (new Table 1 and Fig. 7), as well as new Table A1 for the Appendix, presenting recalibrated phytoplankton constants; 3) something that we call Clarifying considerations (referred to as CC) that contain some material (maps, pictures, etc.) to which we refer to- but still do not intend copying it into the manuscript.

Reviewer: 1

This paper presents a largely theoretical physical-biogeochemical simulation of Lake Onego for a period of 40 years. The paper seeks to use a modelling approach to collate some of the relatively sparse and disparate sources of information available on the lake. While this approach is commendable, it needs to be well supported with a sound underpinning modelling framework. Such a framework would involve: - Being sure to collate the available and relevant sources of information available on the lake. This was not done adequately in my opinion and several of the papers that were part of a special issue on Lake Onego (Inland Waters Vol 9, Issue 2), and contained relevant information, were not cited, while at the same time the authors stated that "there is almost no empirical information on the major biogeochemical variables and fluxes".

The authors have partially addressed this point – noting the inclusion of the Efremova et al. 2019 paper (as cited by the authors).

Unfortunately, we should consider as questionable the Pmin values (ranges/mean) in the open waters in 2016 presented in Table 7 (Efremofa et al., 2019) as (1-1)/1 ug L-1, because, according to corresponding manuals of 2006 and 2019, the standard methods of analysis they used (their Table 2) have a 5 ug L-1 limit of detection. Though, the reported TP values are more reliable.

We have also used from that issue the paper by Filatov et al. (2019) with the satellite information on ice coverage. Otherwise, the problem with this large international Project in relation to our

study is that however interesting and important its achievements are, there are no information that could be directly used, for instance, for the data-model comparison. The paper by Suarez et al., 2019, to which we also refer at L277, deals with phenomena and mechanisms at a diurnal scale that is in a time-space window beyond simulated daily step. The estimates of Chl-a reported in this paper from a few profiles measured in mid-March and June 2017 at two specific locations, on the one hand, cannot be directly compared to simulated biomass even as "typical values" because of unknown C:Chl ratio (see also considerations about satellite information below). On the other hand, as stated by the Suarez et al., the reported values are not unusual for the location and situation and, to our mind, are covered enough by the general statement al L277 "… were in good agreement with the phenology of the Lake Onego ecosystem deduced from field observations". We could have compared the sedimentation velocities with our simulated velocities, but those were not measured but rather estimated from some semi-theoretical Stokes-like equation, similar to that used in our model.

Another paper from this Project by Perga et al. (2021) (Fasting or feeding: A planktonic food web under lake ice, DOI: 10.1111/fwb.13661) studied mechanisms acting at the individuum's level and also does not have direct relation to the simulated lumped variable "Zooplankton", while the zooplankton data are presented as abundances and % taxonomic contribution, that are also incomparable to our simulated biomasses. Although we could add the reference to Perga et al. for completeness of coverage at L277 about typical phenology, according to two phrases from their Abstract: "The algal biomass was low under ice and mostly dominated by large diatoms" and "Environmental conditions under the ice of Lake Onego do not depart significantly from those observed in lakes of similar latitudes."

- A sound modelling process involves calibration and validation against concentrations and/or biogeochemical fluxes. The comparison of temperature made by the authors: "Taking this expected bias into consideration, the simulated water temperature (11.85 ± 3.92 °C; median 13.15 °C) matches the observations (13.05 ± 4.82 °C, median 14.40 °C) well" is inadequate and direct comparisons (observations vs. simulation output) should have been made together with the relevant underpinning errors statistics (e.g., R2, RMSE, PBIAS, etc.).

The authors have not in my opinion adequately addressed this point. They suggest that "the simulated water temperature (11.85 ± 3.92 °C; median 13.15 °C) matches the observations (13.05 ± 4.82 °C, median 14.40 °C) rather well" but the reader is not told what the error statistic (±) relates to and it is not clear why model output was not aligned with comparable location of stations in claiming that there was bias due to 70% of measurements coming from coastal areas and bays.

Here, we start with repeating our explanations about thermo-hydrodynamics also given to Reviwer 1.

First of all, as explained on L161-166 we had no intention to thoroughly validate the simulated hydrophysics, for which there are more publications but which is quite a different separate task, and focused instead on those features that are most important for the seasonal dynamics of biogeochemical cycling and could be compared to some available estimates based on observations.

We also had in mind that 30-year atmospheric forcing based on the ERA-Interim reanalysis fields (https://www.ecmwf.int) with a basic spatial resolution of 80 km (based on a comparatively sparser network of meteorological stations in this region) is hardly suitable for the reliable sequential reproduction of transient synoptic situations of about 5-7 day duration made day-by-day for 30 years. Therefore, we did not even attempt the detailed pair-wise comparison of 532 irregularly scattered measurements of the surface temperature made in June–August over 15 years of 1992–2007 (L180), *a priory* considering it confusing and misleading rather than validating. Instead, we estimated average summer temperatures as indication of the general plausibility of the simulated summer thermic situation. For clarity, we'll revise the description as "… the simulated water temperature (**mean ± standard deviation**, 11.85 ± 3.92 °C; median 13.15 °C) matches the observations (13.05 ± 4.82 °C, median 14.40 °C) …". As for the aligning of sampling locations made over 15 years here and there, we were sure in its futility for the reason given above and still are afraid that such picture and corresponding considerations would make the main biogeochemical story unnecessarily complicated and unfocused.

It looks from Fig. 5 like a comparison was done for three specific stations, but the R values are rather poor for temperature and a 1:1 line should also be shown to examine if there was systematic bias.

Fig. 5 is prepared in addition to estimates of the summer surface temperature over the entire lake area. Of course, the surface water temperatures, occasionally measured here and now, is highly influenced by the real synoptic atmospheric variations and internal small-scale water movements (cf. L187-191). On the other hand, the seasonal dynamics of the surface water temperature reflects a complex development of the convective spring and autumn turnover with a formation of summer stratification in between (see Fig. 9a). That's why we consider a steady reproduction from year to year of such dynamics of spring warming and autumn cooling in a 30-year simulation at both the entire lake scale (L179-185) and for the three offshore stations (Fig. 5) as indicator of its plausibility, while R values calculated at such poor irregular observations (cf. amount and location of points in both panels) as very high. Some deviation from 1:1 line is clearly seen in the graphs and is partly discussed in the text but, generally, can hardly be used for any well-founded and far reaching conclusions exactly because of the extreme paucity of occasional measurements. Such paucity would also make estimates of RMSE, PBIAS and similar statistics hardly significant at all, and would not add much to the main conclusion about plausibility of the simulated long-term thermo-hydrodynamics.

I note that the authors have included a figure (Fig. 4) showing observed and simulated ice cover but give no statistics for goodness of fit and no detail about the way in which ice cover was simulated.

The ice cover was simulated with the package included in MITgcm model (L101-102), which simulates a percent of the ice coverage of a grid cell and has already been successfully used for large boreal lakes (L161-163), including initial test runs for Lake Onego. That's why we have not tuned (calibrated) the original ice model parameterizations and just demonstrated its good performance at Lake Onego with Fig. 4. Statistically, R=0.99 and RMSE = 6.3% On the other hand, calculation of the mean and standard deviation for the seasonal variation from 0 to 100% has hardly much of interest. Correspondingly, we'll expand the text at L171 with the following:

**"… seasonal ice phases rather accurately (Fig. 4), with the coefficient of linear correlation R=0.99 and RMSE = 6.3%. Most closely coincide the onset of freezing, observed from mid-December to early January, and the final ice clearing occurring from mid to late May. Calculation of the mean and standard deviation for the seasonal variation from 0 to 100% has hardly much of interest."**

Dispute biasing the general level of details in MITgsm description, we could revise L101-102 as follows: **"… the SeaIce package included in the MITgcm model complex was used to simulate a percent of the ice coverage of grid cells"**.

Figure 6 caption should explain the black and blue lines and give the temperature for the 'biological summer'.

We'll edit Fig. 6 caption as follows:

**Figure 6: Simulated duration of the period of "biological summer" (shaded area delimited by the start and end day of Gregorian Year, black curves) and its average surface water temperature (blue curve) with estimated trends (black and blue lines, respectively)**

Most importantly, the reader has no information if vertical thermal stratification is captured in the model; are there really no vertical profiles of temperature?

We'll add new Fig. 6 (see below) after the present Fig. 5 with the following text, to which we'll appropriately refer also somewhere at 370. Together with Fig. 6 we'll add the following text:

**"As shows a comparison between average vertical profiles of the water temperature, simulated and reconstructed from measurements (Fig. 6, cf. also Fig. 11a), the summer thermic vertical structure is also simulated rather reasonably. Unfortunately, the further statistical analysis of model-data comparability is prevented by both paucity and irregularity of observations scattered over two decades with some years missed entirely".**

I appreciate that data may have been sparse but there are methods to counter this, e.g., use of remote sensing to provide optically active surface water constituents that can be compared with model output. Sensitivity analysis is also another useful approach to develop confidence in the model simulations and be able to define a range of output as part of a model error analysis. Bootstrapping approaches are also useful for sparse data. Without this the model simulations become a largely theoretical exercise because we do not know the accuracy of the simulation output.

I didn't see a lot of additional effort to address suggestions around sensitivity analysis or the use of remote sensing data to demonstrate spatial variability in the model.

First, we'd like to repeat our explanations given to similar questions by Reviewer 1.

About sensitivity analysis. While adapting SPBEM to Lake Onego, we have made several test and calibration runs, which is the usual practice and hardly worth mentioning in the text but which are not suitable for the uncertainty estimates in the sense indicated by you. Also, we are pretty familiar with- and confident in the biogeochemical module behavior from over two decades of its exploitation (appropriately referred to in the different sections of the text), as

hopefully, are many our readers. Therefore, the formal sensitivity analysis has never been considered as a goal of this study and we have not tried to artificially and unnecessarily alter constants, the least parameterizations. Also, we intentionally used here the word "plausibility" (rather in a sense of G. Polya "Mathematics and plausible reasoning") instead of stronger and more certain "reliable", "realistic", "accurate" and such (again L157-158).

About satellite information. The remote sensing could, in general, be used for validation of some environmental characteristics (water colour, suspended matter, chlorophyll, etc.). But we would not use it, at least, for two reasons. Fundamentally, however important are the synchronous chlorophyll fields obtained by remote sensing as a ***relative measure*** of phytoplankton biomass and its dynamics (e.g. Kahru et al., 2016), they cannot be reliably used for estimation of in situ phytoplankton biomass without special empirical conversions, localized both in space and time, accounting for an order of magnitude seasonal variation of C:Chl ratio, from about 20-30 in spring to 200-300 g C : g Chl-a in summer (cf. Meier et al., 2018 and references therein). Furthermore, in contrast to Lake Ladoga (see Pozdnyakov and Filatov. Interannual water quality variations in Lake Ladoga in spring during 2016 and 2017: satellite observations. Fundamentalnaya i Prikladnaya Gidrofizika. 2021, 14, 1, 79–85. doi: 10.7868/S2073667321010081 (in English); Morozov et al., "Long-term phenology of water quality parameters in Lake Ladoga: A satellite-based study", submitted to JGLR), the specific retrieval algorithm for Lake Onego chlorophyll has not been constructed yet. Some pilot attempts (CC6) could hardly be used for any reliable analysis, the least, validation.

On line 215-216, it is stated that "Presented concentrations of inorganic and total nutrients were averaged over the whole water body, from the surface to the bottom". I understand why this might be done for a winter mixed period, but not for summer. I'm unclear on the following sentence as well, and it seems to be that distributing DIP through the water column in summer would lead to some large inaccuracies. Indeed, earlier it is stated (line 211) that "...variables averaged over the entire Lake Onego"; was this a volumetric average or was it a vertical average for the water column? The authors seem to add doubt between observations and model assumptions with the statement that: "Consequently [because the authors chose to average nutrient concentrations over the whole water body], the dissolved inorganic phosphorus (DIP), comprising summer phosphorus accumulation in the hypolimnion, was never fully depleted'.

We agree that our imprecise formulations with respect to different presented and discussed scales (namely, seasonal to long-term interannual in Section 3.1.2 vs. mainly seasonal in Section 3.2) could cause some confusion. According to L211, these are lake-wide volumetric average concentrations needed to characterize seasonal and interannual dynamics of total amounts (e.g. L217-219) as a large-scale indicator of the trophic state responding to external nutrient inputs. Also, indication "…from the surface to the bottom…" might be confusing by hinting at the water column, i.e. under sq. m and we'll delete it. Indeed, the time-depth variations, showing also summer phosphorus accumulation in the hypolimnion (see also CC 8), are presented in our Fig. 9b, which will be referenced here as well. Correspondingly, we'll revise L215-219 as follows:

**"Presented concentrations of inorganic and total nutrients were averaged over the whole water body. Consequently, the dissolved inorganic phosphorus (DIP) content, comprising summer phosphorus accumulation in the hypolimnion (see Fig. 9b below), was never fully depleted. The lake-wide averaged winter maximum values (Fig. 7 c, e) multiplied by the Lake Onego model volume (297 km3) could be used to conveniently estimate total nutrient**

**stocks as a large-scale indicator of the lake's trophic state responding to variations in external nutrient inputs."**

- Almost no information is given on the parameters that go into the model. Parameters like sediment nutrient release rates, deoxygenation rates, algal growth rates, etc., need to be provided in any modelling exercise; they serve as a basis for future work and refinement (e.g., in experimental work) and they should generally fit within literature ranges.

Information on parameters is still not given.

Unfortunately, we again do not quite understand the question – is it about a) model formulation or b) resulting simulated rates and fluxes? If "a", then the full model description is given at many pages in the Open Access paper by Isaev et al. (2020), while an adaptive update will be added as Table A1. Consequently, everyone can estimate resulting rates at the temperatures and concentrations of his/her interest, resulting in the endless amount of potential permutations. If "b", then we completely agree about the current and future role of out model as a helping tool in studies of Lake Onego ecosystem and will more precisely formulate it in Introduction (see below). Correspondingly, the entire paper is dedicated to presentation of some simulated rates, which we selected considering the total paper volume. We would not repeat the text here, but just for example, please, pay attention to the simulated primary production in Figs 7a and 8a and 8d and Table 2-3, as well as new Table 1 and Fig. 10, that will be compared compared to observations at L320-350. The simulated oxygen consumption is exemplified at L386-389. The entire Section 3.3 is about simulated fluxes, including Table 4, where presented annual fluxes (e.g. nutrient uptake and limnic recycling, burial and sedimentation, nutrient release and denitrification) could be converted to area or volume units by the lake's area and volume. More simulated rates are exampled and compared to literature ranges at L440-465.

A hint of parameterization is given in the sediment N and P content values given in Table 1 but it is notable that the model sediment N content is mostly greater than the range given for the measured values – and this is mostly the case for sediment P also. This raises some major question marks about the N and P mass balances that are given for the lake.

The parameterization of sediment variables and fluxes is given in Isaev et al. (2020; Table 1; Eqs. 15-16; A38-39; A55-64; Table A2-A3), while Table 1 in this manuscript contains simulated values for comparison to our compilation of some measurements (see CC2-5), which gives a rather good comparability (with revised assumptions). Please, check the actual version of Table 1.

There is a great deal of uncertainty for the reader relating to the way in which the biogeochemical model was calibrated. The paragraph in the Ecosystem variables section (lines 211 to 222) did not alleviate these concerns.

The paragraph at L211-222 is not as much about calibration but rather explains unit conversions necessary for intercomparing model variables and fluxes with typically reported from measurements. Here we repeat explanations about calibration given to Reviewer 1.

We have both explanations and additions to the text related to calibration.

First of all, we'd like to stress that the biogeochemical module has been extensively calibrated within BALTSEM model, plausibly reproducing ecosystem dynamics in the entire Baltic Sea (e.g. Meier et al., 2018), from the cold, annually ice-covered, almost fresh, and severely P-limited Bothnian Bay (i.e. very much Onego-like), to the warmer, mesotrophic Gulf of Finland and the Kattegat with a single set of parameterizations and constants in both basin-wise horizontally averaged and true 3D versions (e.g. Gustafson et al., 2017; Ryabchenko et al., 2016, Isaev et al., 2020). Besides, similar formulations have already been favorably tested at Lake Ladoga (Isaev and Savchuk, 2020). Such simultaneous coverage of a wide range of ecological conditions makes us somewhat confident in application of largely the same set of formulations to Lake Onego.

Unfortunately, the presented manuscript creates a wrong impression that we fully avoided the calibration. As can be seen from SPBEM formulation (Table A1 in Isaev et al., 2020), the major difference between "cyanobacteria" and "summer species", given in parameterization, is the capability to fix molecular nitrogen under appropriate conditions. Without such conditions both variables behave almost identical as was known from the Bothnian Bay and Lake Ladoga simulations and was demonstrated by the initial runs for Lake Onego. Therefore, as indicated in the text (lines 87-93), we excluded the "diazotrophic cyanobacteria" group as a separate variable, thus actually merging such other ecosystem functions and biogeochemical fluxes as nutrient uptake, mortality, sinking, etc., into "non-diatoms" group. Such adaptation requested recalibration necessary also to better separate dynamics of cold-water diatoms from summer "non-diatoms". As was also shown by the initial test runs, all the other temperature- and concentration dependent processes, being already calibrated for similar conditions in the Baltic Sea and Lake Ladoga, have not requested urgent re-calibration, unsupported by sufficient amount of contradicting reliable observations, and were left as they were. Based on these considerations we will revise the text situated at lines 90-93, as follows (and add a Table A1 in Appendix, if requested by the Editor):

**"Thus, autotrophs were presented by only two variables, diatoms and non-diatoms, that comprised all the other (summer) phytoplankton species, for example, chlorophytes, chrysophytes, and cyanobacteria. Such reformulation requested recalibration of several phytoplankton parameters, necessary also to better separate dynamics of summer "non-diatoms" from cold-water diatoms (Table A1). As was also shown by several test runs, all the other formulations, being extensively calibrated and tested in similar temperature and trophic conditions (e.g. Gustafsson et al., 2014; Isaev et al., 2020 and references therein) have not requested further fine-tuning in the absence of abundant reliable contradicting observations."**

We also suggest to add Table A1, in order to update a set of constants from Isaev et al., 2020 (with actual values used for Lake Onego)

- The reader is given no overview of measured forcing data inputs into the model. For example, the 'estimated' river runoff and N and P loads (Fig. 3) should have had the observations included also. Further, the meteorological variable inputs to the model should have been clearly specified.

There were quite a few typographical errors through the paper (e.g., spelling mistakes in Fig. 1) that would need to be corrected in future iterations of this paper.

The authors now provide a brief description of inflow, outflow and meteorological forcing data for the model input.

See further explanations below

I would suggest not mixing the results and discussion into a single section, which would help to add clarity.

Not done. This paper could easily be improved by separation of Results and Discussion sections.

Unfortunately, we have always had difficulties with such separation for the modelling papers in general and, particularly, with this one, when so little observations are available. In papers presenting field and laboratory studies, the presentation and description of the collected data could be comparatively cleanly separated from the assessment and interpretation of the obtained data with further discussion of the knowledge acquired. In our manuscript, a simple succession of Figures and Tables, collected in Results, would inevitably cause a lot of repetition in Discussion, including unnecessary and inconvenient sending the reader back and forth between these two parts. That is why we still prefer (L150-158) to consecutively present different aspects of simulated nutrient biogeochemistry (reanalysis) of Lake Onego ecosystem (long-term development, spatial distribution, seasonal dynamics, nutrient fluxes and budgets), on the way comparing them to available sparse information.

In other instances, elements of Introduction should not appear in the Methods. The Introduction would benefit from a clearly defined scientific hypothesis or test being put forward (in the last paragraph of the Introduction), i.e., this would form the basis of the model testing or scenario generation.

No clear scientific test or hypothesis presented. What was the purpose or objective of this paper?

In a possible revision, we'll extent and finish Introduction with the following:

**"Thus, the main purpose of this paper is a presentation of the 3D ecosystem model capable to a certain extent fill the historical deficit in observations of nutrient variables and, especially estimates of the biogeochemical fluxes. According to one of the major functions of simulation modelling, we intend implementing this model as a complimentary form of studies of Lake Onego ecosystem, providing a unifying formal platform for testing and discussing consistency of both model parameterizations and results of hydrological, hydrochemical, hydrobiological, and geochemical research. Furthermore, the model will be implemented as a major tool for a wide range of projections, from applied tasks of localization of fish farms, water intakes, and wastewater outlets to long-term large-scale ecosystem evolution under different scenarios of climate change and socio-economic development."**

What efforts were made to validate the discharge and nutrient concentrations measurements? It is not clear why the reviewer cannot see the interpolated load values (Fig. 3) plotted against the actual measurements (as points), so that the reader can see the frequency of measurements. Were

the so-called "upward tendency" and "distinctly decreased" changes in loads with time actually significant? How were concentrations of dissolved nutrients in the inflows determined?

We cannot "validate" the forcing functions *reconstructed* from the published sources. That is why there are no "actual measurements", additional to referred sources and explained at L118-132. River runoff is reconstructed from the water balance and further split between 13 rivers in (Filatov, 2020), while nutrient concentrations were reconstructed from just three sets of surveys with asynchronous sampling made in 1985-1986, 2001-2002, and 2007-2008. Another similar survey was made in 2020 and was just published by Galakhina et al. (2021) (see below).

We further suggest a bit more details of the input reconstruction, just to stress that it is neither expedient nor even possible to present some "observed/measured" points. The suggestions for revision are:

At L119 "**The total *RECONSTRUCTED* water discharge (Fig. 3a)…**"

L126-130 should be revised as "**Sabylina (2016) compiled information on the total content and inorganic fractions of nutrients in river waters that was obtained in surveys performed in 1985-1986, 2001-2002, and 2007-2008. Assuming the similarity of concentrations in rivers that drain catchments with similar landscapes, Sabylina (2016) provided reasoning for the aggregation of many streams and rivers into larger units flowing into Lake Onego**. **Correspondingly, we multiplied these concentrations by the reconstructed monthly water discharge for the indicated 13 rivers, filling the gaps over years without observations by linear interpolation between available concentration values.**"

Formal estimates of significance of such multi-step composite reconstructions are hardly expedient, while the drawn regression lines should just help to better discern tendencies seen by the naked eye. However, we'll add further confirmation at L230:

"**As shows comparison between surveys of 2007-2008 (Sabylina, 2016) and 2019-2020 (Galkhina et al., 2021), these tendencies are continuing. In result, the *prescribed* bioavailable…**"

No formal calibration and validation of the model is carried out by the authors, raising doubts about the predictive capabilities of the model and more effort (e.g., remote sensing) should have gone into supporting this process.

Here we again would like repeating our explanations about calibration and validation given to Reviewer 1, while our considerations of the remote sensing support are given above.

First of all, we'd like to stress that the biogeochemical module has been extensively calibrated within BALTSEM model, plausibly reproducing ecosystem dynamics in the entire Baltic Sea (e.g. Meier et al., 2018), from the cold, annually ice-covered, almost fresh, and severely P-limited Bothnian Bay (i.e. very much Onego-like), to the warmer, mesotrophic Gulf of Finland and the Kattegat with a single set of parameterizations and constants in both basin-wise horizontally averaged and true 3D versions (Gustafson et al., 2017; Ryabchenko et al., 2016, Isaev et al., 2020, referred to in the manuscript). Besides, similar formulations have already been favorably tested at Lake Ladoga (Isaev and Savchuk, 2020). Such simultaneous coverage of a

wide range of ecological conditions makes us somewhat confident in application of largely the same set of formulations to Lake Onego.

Unfortunately, the presented manuscript creates a wrong impression that we fully avoided the calibration. As can be seen from SPBEM formulation (Table A1 in Isaev et al., 2020), the major difference between "cyanobacteria" and "summer species", given in parameterization, is the capability to fix molecular nitrogen under appropriate conditions. Without such conditions both variables behave almost identical as was known from the Bothnian Bay and Lake Ladoga simulations and was demonstrated by the initial runs for Lake Onego. Therefore, as indicated in the text (lines 87-93), we excluded the "diazotrophic cyanobacteria" group as a separate variable, thus actually merging such other ecosystem functions and biogeochemical fluxes as nutrient uptake, mortality, sinking, etc., into "non-diatoms" group. Such adaptation requested recalibration necessary also to better separate dynamics of cold-water diatoms from summer "non-diatoms". As was also shown by the initial test runs, all the other temperature- and concentration dependent processes, being already calibrated for similar conditions in the Baltic Sea and Lake Ladoga, have not requested urgent re-calibration, unsupported by abundant contradicting reliable observations, and were left as they were. Based on these considerations we will revise the text situated at lines 90-93, as follows (and add a Table A1 in Appendix, if requested by the Editor):

**"Thus, autotrophs were presented by only two variables, diatoms and non-diatoms, that comprised all the other (summer) phytoplankton species, for example, chlorophytes, chrysophytes, and cyanobacteria. Such reformulation requested recalibration of several phytoplankton parameters, necessary also to better separate dynamics of summer "non-diatoms" from cold-water diatoms (Table A1). As was also shown by several test runs, all the other formulations, being extensively calibrated and tested in similar temperature and trophic conditions (e.g. Gustafsson et al., 2014; Isaev et al., 2020 and references therein) have not requested further fine-tuning in the absence of abundant reliable contradicting observations."**

About validation

Appropriate model validation has always been our concern for decades (e.g. Savchuk and Wulff, 1996; Savchuk, 2002; Savchuk et al., 2012, Gustafsson et al., 2017; Isaev et al., 2020), especially with a wealth of data available for the Baltic Sea. Unfortunately, the situation is drastically different with Lake Onego.

Our mechanistic model is based on a mass balance approach, describes internal biogeochemical cycles and accounts for external sources and sinks (imports and exports), either prescribed as forcing functions or computed according to formulations. Consequently, both all the simulated fluxes and concentrations resulting from their interplay are strongly deterministically coupled and thus, confined. Therefore, their reliability should be judged by a *simultaneous* fitting of many fluxes and concentrations in the known ranges reported for both Onego and similar boreal oligotrophic lakes. For example (our Table 3), the nutrient sedimentation of OM cannot be very much higher than nutrient uptake during primary production of OM simulated with the plausibly

given phytoplankton specific growth rates (note good PP validation, new Table 1 and Fig. 10 below). Similarly, the sediment release (and denitrification) of nutrients cannot be order of magnitude, or even several-fold higher or lower, thus causing (unreported? unobserved?) fast accumulation or depletion of sediment nutrients. The plausibility of simulated rates is estimated by a comparison to sediment rates from similar environments (our lines 443-465). This can be said about all the other processes in Fig. 1 and Table 3.

Initially, the main approach was to compare simulation to information already published as tables, graphs, and maps in books, atlases, and papers (appropriately referred to in the manuscript), thus leaving responsibility of interpretation of scarce data to corresponding authors-experts. Besides, following the usual practice of substantiating our own judgments and statements, including numbers, we tried to give appropriate references to all the quantitative and qualitative information, which we used for comparison, trusting the sources and expecting similar trust from the readers to our compilations and references. However, your request made us to dig up some information, just to indicate that there are only a few dozen of measurements irregularly scattered over two decades, mostly only in summer (see new Table 1and Fig. 6 and A10 below). Therefore, we prefer to talk about typical ranges rather than calculate some highly uncertain (or even mathematically incorrect) statistics, as we already explicitly admitted at L157-158.

A recent paper by Galakhina et al. "Current chemistry of Lake Onego and its spatial and temporal changes for the last three decades with special reference to nutrient concentrations" just pre-printed in "Environmental Nanotechnology, Monitoring and Management" (https://doi.org/10.1016/j.enmm.2021.100619) presents results of 3 (three) surveys in September 2019, June, and August 2020 (that is beyond our simulation interval) at 35 stations in different regions of Lake Onego as well as refers to some older scattered data from 1992-2018. The samples were taken only from surface and bottom layers. Note, we intend to use this paper and information from it in a possible revision of our manuscript at L 230:

**"Such simulated quasi-stability of TP concentrations and clearly decreasing DIN concentrations (Fig. 8 c, e)) is validated by the recent field surveys. Galakhina et al. (2021) found at the surface of pelagic part of Lake Onego a statistically significant decreasing trend of the DIN:TP weight ratio from 33.7 in 1992-1995 to 23.7 estimated from the field surveys made in September 2019, June and August 2020, which is well comparable to a simulated decrease of DIN:TP ratio from $36.5 \pm 1.9$ in 1992-1995 to $21.2 \pm 1.1$ in 2011-2015; these mean $\pm$ S.D. values are computed for the surface layer in I-VI limnic areas (cf. Fig. 2) over biological summer (cf. Figs. 7 and 11 a)."**

It is not clear what the authors are pointing out in the following: "Taking into account all the uncertainties of such comparisons, starting with measurements being made over years from the mosaic distribution of real sediment types, and ending up with simplified and spatially invariable sediment parameterizations disregarding sediment types, the simulated areal concentrations could be considered plausible". It is not clear what would be plausible or implausible.

We agree, the sentence is kind of clumsy and cumbersome. In a possible revision, we'll replace it with the following additional explanations:

**Wide ranges of published sediment characteristics compiled in Table 2 from maps in (Filatov, 2010, pp. 100-101) and a few numbers in (Kalinkina and Belkina, 2018, Table 1), are based on samples taken over many years from the mosaic distribution of real sediment types, thus  guaranteeing the exact positioning, sufficient homogeneity, and reproducibility of measurements at indicated locations. On the other hand, our simplified and spatially invariable sediment parameterizations totally disregard the diversity of sediment types, while registering a dynamic balance between the source (sedimentation) and the sinks (nutrient release and burial), and closing nutrient cycles by regeneration (see Table 3 below). Taking into account these interpretations, a comparability between simulated values and estimates, recalculated from measurements, could be considered plausible.**
* * *
New table and figures

[Figure]

New Figure 6. Observed (red) and simulated (blue) average vertical distribution of the water temperature in three limnic areas of Lake Onego (cf. Fig. 2 and (old) 9a), showing also time coverage and amount of observations. Note differences in scales.

New Table 1 –Phytoplankton primary production in Lake Onego according to observations (Rukhovetz and Filatov, 2010, Table 1.4) and simulation for the period from 1989 to 2006, mg C $m^{-2}$ $day^{-1}$.

| | Data | Model | | |
|---|---|---|---|---|
| | summer | May – Oct | Jun - Sep | Jul – Sep |
| Southern Onego | 88.3 ± 15.5 | 82.4 ± 60.7 | 72.0 ± 48.8 | 53.6 ± 4.7 |
| Central Onego | 96.3 ± 10.5 | 121.5 ± 109.7 | 111.2 ± 105.2 | 69.0 ± 15.4 |
| Petrozavodsk Bay | | | | |
| top part | 412.9±62.7 | 350±121 | 491±131 | 494±86 |

| | | | | |
|---|---|---|---|---|
| central part | 199.8±38.3 | 254±56 | 297±61 | 284±44 |
| outer part | 122.3±21.7 | 180±57 | 166±37 | 151±25 |
| Kondopoga Bay | | | | |
| central part | 286.7±24.2 | 221±90 | 324±93 | 288±76 |
| outer part | 217.4±23.3 | 153±71 | 185±81 | 134±26 |

[Figure]

New Fig. 10. Model-data box-and-whisker plots of the phytoplankton primary production (mg C m$^{-2}$ day$^{-1}$) in different limnic areas of Lake Onego (cf. Fig. 2). Note paucity and irregularity of observations.

Suggested Appendix

**Table A1.** Parameters for the autotroph groups.

| Symbol | Parameters | Units | Diatoms | NonDiatoms |
|---|---|---|---|---|
| $\lambda_{AN}$ | N/P ratio | mgN/mgP | 7 | 7 |
| $a_{gi}$ | Maximum growth rate at 0 °C | day$^{-1}$ | 1.25 | 0.75 |
| $b_{gi}$ | Temperature constant for growth and mortality | °C$^{-1}$ | 0.078 | 0.12 |
| $I_{0i}$ | Optimal photosynthetically active radiation | W m$^{-2}$ | 25 | 50 |
| $h_{Ni}$ | Half-saturation constant for inorganic nitrogen | mg N m$^{-3}$ | 7.0 | 3.5 |
| $h_{Pi}$ | Half-saturation constant for phosphate | mg P m$^{-3}$ | 1.5 | 1.5 |
| $c_{rN}$ | Threshold ammonium concentration | mg N m$^{-3}$ | 21.0 | 21.0 |
| $a_{mi}$ | Mortality rate at 0 °C | day$^{-1}$ | 0.4 | 0.15 |
| $b_{mi}$ | Temperature constant for mortality | °C$^{-1}$ | 0.063 | -0.2 |
| $a_{si}$ | Sinking velocity at 0 °C | m day$^{-1}$ | 0.5 | 0.1 |
| $\gamma_{mi}$ | Mortality rate adjustment for "fit" conditions | | 4 | 4 |
| $\gamma_{si}$ | Sinking velocity adjustment for "fit" conditions | | 4 | 4 |
| $\alpha_i$ | Availability as food source | | 1.0 | 1.0 |

Clarifying considerations (CC)

CC1. Availability of the phytoplankton primary production measurements in the different limnic areas of Lake Onego. We show generalization of this information in new Fig. 10

| | | Number of measurements | | | | | |
|---|---|---|---|---|---|---|---|
| | | Month | | | | | |
| Region | Years | Jun | Jul | Aug | Sep | Oct | All |
| BigOnego | 1989 - 2010 | 8 | 5 | 9 | 3 | 3 | 28 |

| | | | | | | |
|---|---|---|---|---|---|---|
| CentralOnego | 1994 - 2010 | 17 | 6 | 13 | – | – | 36 |
| SouthOnego | 1994 - 2005 | – | 3 | 5 | – | – | 13 |
| PetrozavodskBay | 1989 - 2010 | 27 | 11 | 13 | 6 | 3 | 60 |
| KondopogaBay | 1989 - 2010 | 22 | 12 | 34 | 14 | 11 | 93 |

CC2. Distribution of the sediment types (granulometric composition at p. 100 in Filatov (2010). (left)

[Figure]

[Figure]

CC3. Distribution of the C-org (left), TP (middle) and N-org (right) in the Onego sediments at p. 101 (Filatov, 2010). (right)

Калинкина Н. М., Белкина Н. А. Динамика состояния бентосных сообществ и химического состава донных отложений Онежского озера в условиях действия антропогенных и природных факторов // Принципы экологии. 2018. № 2. С. 56–74. DOI: 10.15393/j1.art.2018.7643

[Figure]

Рис. 1. Расположение станций отбора проб ДО и бентоса. 1 – зона интенсивного антропогенного воздействия; 2 – буферные зоны заливов; 3 – глубоководные участки

Fig. 1. Location of sampling stations of bottom sediments and benthos. 1 – zone of intensive anthropogenic impact; 2 – buffer zones of the bays; 3 – deep water areas

CC4. Sampling sites in (Kalinkina and Belkina, 2018)

Калинкина Н. М., Белкина Н. А. Динамика состояния бентосных сообществ и химического состава донных отложений Онежского озера в условиях действия антропогенных и природных факторов // Принципы экологии. 2018. № 2. С. 56—74. DOI: 10.15393/j1.art.2018.7643

Таблица 1. Химический состав донных отложений Онежского озера (слой 0-5 см)

| Период | № ст. | Eh, мВ | pH | C | ППП | N-NH₄ | N общ | Fe | Mn | K вал | P вал |
|---|---|---|---|---|---|---|---|---|---|---|---|
| | | | | | | | % от сухой навески | | | | |
| *Кондопожская губа* | | | | | | | | | | | |
| 1991-1999 | К_3 | 89 | 6.53 | 19.8 | 45 | 0.228 | 0.70 | 1.6 | 0.34 | 0.13 | 0.21 |
| 2000-2005 | | 11 | 6.23 | 16.8 | 42 | 0.010 | 0.63 | 3.1 | 0.19 | 0.14 | 0.19 |
| 2005-2010 | | 282 | 6.19 | 26.8 | 56 | 0.048 | 1.01 | 3.1 | 0.28 | 0.18 | 0.27 |
| 1991-2000 | К_4 | 60 | 6.80 | 5.1 | 9 | 0.007 | 0.51 | 0.8 | 0.09 | 0.05 | 0.10 |
| 2000-2005 | | 68 | 6.88 | 13.1 | 25 | 0.057 | 0.47 | 3.1 | 0.27 | 0.11 | 0.16 |
| 2005-2010 | | 77 | 6.57 | 12.4 | 33 | 0.073 | 0.60 | 3.6 | 0.14 | 0.15 | 0.21 |
| 2010-2015 | | 138 | 6.24 | 13.7 | 29 | 0.008 | 0.61 | 3.0 | 0.17 | 0.15 | 0.20 |
| 1991-1995 | К_6 | 151 | 6.77 | 7.2 | 27 | 0.046 | 0.72 | 3.2 | 1.08 | 0.18 | 0.27 |
| 2000-2005 | | 247 | 6.67 | 7.1 | 21 | 0.063 | 0.48 | 7.0 | 1.07 | 0.25 | 0.31 |
| 2005-2010 | | 73 | 6.73 | 7.2 | 18 | 0.004 | 0.20 | 6.6 | 0.47 | 0.24 | 0.28 |
| 1990-1995 | К_7 | 301 | 6.57 | 2.6 | 14 | 0.022 | 0.62 | 0.6 | 0.43 | 0.07 | 0.13 |
| 2001-2005 | | 366 | 6.44 | 4.4 | 17 | 0.079 | 0.30 | 5.6 | 0.99 | 0.15 | 0.22 |
| 2005-2010 | | 374 | 6.32 | 4.4 | 10 | 0.003 | 0.15 | 5.1 | 1.00 | 0.13 | 0.21 |
| *Петрозаводская губа* | | | | | | | | | | | |
| 1990-1995 | Р_5 | 112 | 6.59 | 1.7 | 4 | 0.020 | 0.20 | 0.9 | 0.06 | 0.06 | 0.11 |
| 2000-2005 | | 290 | 6.99 | 3.5 | 9 | 0.018 | 0.19 | 3.5 | 0.36 | 0.13 | 0.15 |
| 2005-2010 | | 335 | 5.63 | 3.8 | 10 | 0.002 | 0.11 | 4.2 | 0.23 | 0.15 | 0.16 |
| 1990-1995 | Р_2 | 179 | 6.59 | 2.8 | 13 | 0.021 | 0.33 | 1.2 | 0.93 | 0.12 | 0.16 |
| 2000-2005 | | 419 | 6.70 | 3.9 | 9 | 0.019 | 0.31 | 5.6 | 1.19 | 0.15 | 0.20 |
| 2005-2010 | | 582 | 6.54 | 3.8 | 13 | 0.003 | 0.29 | 5.4 | 1.43 | 0.18 | 0.23 |
| 2010-2015 | | 575 | 6.02 | 4.4 | 16 | 0.004 | 0.39 | 5.0 | 1.72 | 0.09 | 0.22 |
| 1990-1995 | Р_3 | 347 | 6.59 | 2.9 | 12 | 0.021 | 0.66 | 1.3 | 1.37 | 0.12 | 0.16 |
| 2000-2005 | | 332 | 6.78 | 4.1 | 12 | 0.020 | 0.26 | 4.7 | 0.48 | 0.16 | 0.21 |
| 2005-2010 | | 353 | 6.08 | 2.6 | 10 | 0.002 | 0.13 | 4.9 | 0.66 | 0.12 | 0.17 |
| 2016* | | 472 | 5.86 | 4.4 | 15 | 0.003 | | | | | |
| *Большое Онего* | | | | | | | | | | | |
| до 2000 | В_1 | 357 | 6.55 | 3.6 | 15 | 0.041 | 0.67 | 1.9 | 1.25 | 0.11 | 0.15 |
| 2000-2005 | | 408 | 6.23 | 4.6 | 16 | 0.023 | 0.28 | 7.3 | 1.48 | 0.15 | 0.22 |
| 2005-2010 | | 429 | 6.46 | 4.2 | 18 | 0.013 | 0.34 | 7.3 | 1.48 | 0.15 | 0.27 |
| 2016 год | | 569 | 6.70 | 4.3 | 17 | 0.008 | 0.57 | | | 0.10 | 0.21 |
| 2000-2005 | В_2 | 341 | 6.14 | 4.9 | 13 | 0.060 | 0.30 | 4.7 | 1.01 | 0.13 | 0.21 |
| 2005-2010 | | 508 | 6.41 | 2.3 | 14 | 0.003 | 0.39 | 5.2 | 0.82 | 0.12 | 0.18 |
| 2016* | | 656 | 6.54 | 4.7 | 18 | 0.003 | 0.50 | | | 0.11 | 0.19 |
| *Центральное Онего* | | | | | | | | | | | |
| 1990-1995 | С_1 | 316 | 6.69 | 3.2 | 17 | 0.039 | 0.55 | 2.5 | 1.81 | 0.15 | 0.31 |
| 2000-2005 | | 515 | 6.41 | 3.5 | 17 | 0.015 | 0.32 | 7.2 | 1.34 | 0.22 | 0.29 |
| 2005-2010 | | 344 | 6.40 | 2.9 | 14 | 0.002 | 0.20 | 7.6 | 0.82 | 0.27 | 0.34 |
| 2010-2017 | | 459 | 6.61 | 2.9 | 13 | 0.003 | 0.30 | 3.4 | 0.98 | 0.08 | 0.28 |

Примечание: * – единичные пробы.

CC5. Table 1. Chemical composition of the Lake Onego bottom sediments (layer 0-5 cm) (Kalinkina and Belkina, 2018)

[Figure]

Май–июнь          Июль

Август          Сентябрь

algal_2_median,[mg/m^3]

0.0   1.0   2.0   3.0   4.0   5.0

CC6. Test interpretations of remotely sensed chlorophyll-a in Lake Onego in 2011

[Figure]

[Figure]

**Fig. 2.** Multiyear seasonal dynamics of the primary production process and abiotic parameters. (*1*) PP, (*2*) surface water temperature, (*3*) PAR, (*4*) N-NO$_3$, (*5*) P$_{total}$.

[Figure]

**Fig. 3.** Multiyear seasonal dynamics of the primary production process and biotic parameters. (*1*) PP, (*2*) diatom phytoplankton biomass, (*3*) biomass of "nondiatom" phytopankton, (*4*) chlorophyll *a*.

CC7. Note that all curves are normalized by a maximum value found on the presented time span and start only in the end of May, when normalized diatom biomass inexplicably goes steeply down from apparently much higher values (from the spring bloom revealed in our simulation, as we interpret the dynamics). Fig.3 also gives a false impression of summer dominance of (3) due to normalization with maximum value, whereas in reality diatoms dominate by biomass over the entire summer

[Figure]

Распределение общего фосфора в поверхностном слое воды летом

Распределение общего фосфора по разрезу А–А, мкг/л

CC8. Summer distribution of TP in the surface layer and cross-section AA in *spring* (*5-11 of June* (sic!)) 1995, summer 2001, and autumn 1992. Note paucity and irregularity of surveys.